# WITH FALSE FRIENDS LIKE THESE, WHO CAN HAVE SELF-KNOWLEDGE?

## ABSTRACT

Adversarial examples arise from excessive sensitivity of a model. Commonly studied adversarial examples are malicious inputs, crafted by an *adversary* from correctly classified examples, to induce misclassification. This paper studies an intriguing, yet far overlooked consequence of the excessive sensitivity, that is, a misclassified example can be easily perturbed to help the model to produce correct output. Such perturbed examples look harmless, but actually can be maliciously utilized by a *false friend* to make the model self-satisfied. Thus we name them *hypocritical examples*. With false friends like these, a poorly performed model could behave like a state-of-the-art one. Once a deployer trusts the hypocritical performance and uses the "well-performed" model in real-world applications, potential security concerns appear even in benign environments. In this paper, we formalize the *hypocritical risk* for the first time and propose a defense method specialized for hypocritical examples by minimizing the tradeoff between natural risk and an upper bound of hypocritical risk. Moreover, our theoretical analysis reveals connections between adversarial risk and hypocritical risk. Extensive experiments verify the theoretical results and the effectiveness of our proposed methods.

## 1 INTRODUCTION

Deep neural networks (DNNs) have achieved breakthroughs in a variety of challenging problems such as image understanding (Krizhevsky et al., 2012), speech recognition (Graves et al., 2013), and automatic game playing (Mnih et al., 2015). Despite these remarkable successes, their pervasive failures in adversarial settings, the phenomenon of *adversarial examples* (Biggio et al., 2013; Szegedy et al., 2014), have attracted significant attention in recent years (Athalye et al., 2018; Carlini et al., 2019; Tramer et al., 2020). Such *small* perturbations on inputs crafted by adversaries are capable of causing well-trained models to make *big* mistakes, which indicates that there is still a large gap between machine and human perception, thus posing potential security concerns for practical machine learning (ML) applications (Kurakin et al., 2016; Qin et al., 2019; Wu et al., 2020b).

An adversarial example is "an input to a ML model that is intentionally designed by an attacker to fool the model into producing an incorrect output" (Goodfellow & Papernot, 2017). Following the definition of adversarial examples on classification problems (Goodfellow et al., 2015; Papernot et al., 2016; Elsayed et al., 2018; Carlini et al., 2019; Zhang et al., 2019; Wang et al., 2020b; Zhang et al., 2020; Tramèr et al., 2020), given a DNN classifier $f$ and a *correctly* classified example $x$ with class label $y$ (i.e., $f(x) = y$), an adversarial example $x_{\text{adv}}$ is generated by perturbing $x$ such that $f(x_{\text{adv}}) \neq y$ and $x_{\text{adv}} \in \mathcal{B}_\epsilon(x)$. The neighborhood $\mathcal{B}_\epsilon(x)$ denotes the set of points within a fixed distance $\epsilon > 0$ of $x$, as measured by some metric (e.g., the $l_p$ distance), so that $x_{\text{adv}}$ is visually the "same" for human observers. Then, an imperfection of the classifier is highlighted by $\mathcal{G}_{\text{adv}} = \text{Acc}(\mathcal{D}) - \text{Acc}(\mathcal{A})$, the performance gap between the accuracy (denoted by $\text{Acc}(\cdot)$) evaluated on clean set sampled from data distribution $\mathcal{D}$ and adversarially perturbed set $\mathcal{A}$.

An adversary could construct such a perturbed set $\mathcal{A}$ that looks no different from $\mathcal{D}$ but can severely degrade the performance of even state-of-the-art DNN models. From direct attacks in the digital space (Goodfellow et al., 2015; Carlini & Wagner, 2017) to robust attacks in the physical world (Kurakin et al., 2016; Xu et al., 2020), from toy classification problems (Chen et al., 2020; Dobriban et al., 2020) to complicated perception tasks (Zhang & Wang, 2019; Wang et al., 2020a), from the high dimensional nature of the input space (Goodfellow et al., 2015; Gilmer et al., 2018) to the

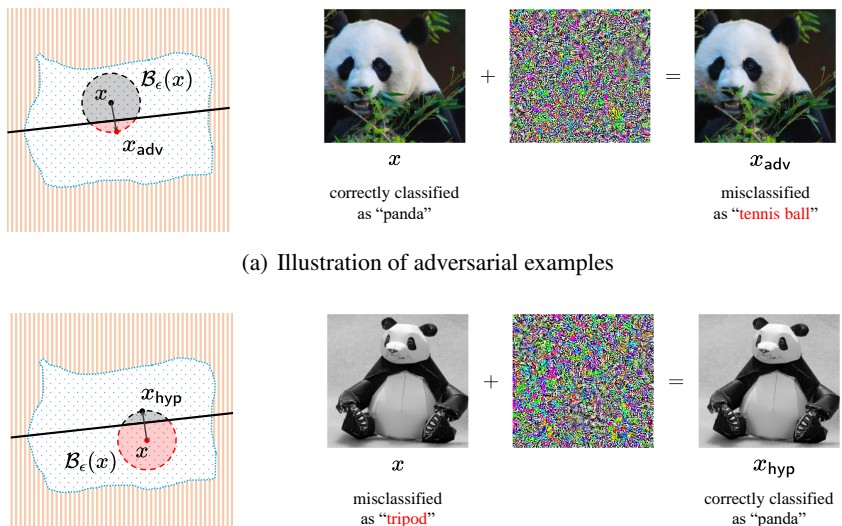

(a) Illustration of adversarial examples

(b) Illustration of hypocritical examples

Figure 1: Comparison between adversarial examples and hypocritical examples. **Left**: Conceptual diagrams for the generation of an adversarial example $x_{\text{adv}}$ and a hypocritical example $x_{\text{hyp}}$. The input space is (ground-truth) classified into the orange lined region (e.g., class "not panda"), and the blue dotted region (e.g., class "panda"). The black solid line is the decision boundary of a non-robust model, which classifies the region above the boundary as "panda" and the region below the boundary as "not panda". Red shadow and black shadow in the ball $\mathcal{B}_\epsilon(x)$ denote that the points in there are misclassified and correctly classified, respectively. As we can see, $x_{\text{adv}}$ or $x_{\text{hyp}}$ can be easily found by perturbing a correctly classified $x$ or a misclassified $x$ across the model's decision boundary. **Right**: A demonstration of adversarial examples and hypocritical examples on real data. Here we choose ResNet50 (He et al., 2016a) trained on ImageNet (Russakovsky et al., 2015) as the victim model. In (a) the correctly classified "panda" can be stealthily perturbed to be misclassified as "tennis ball". In (b) the "panda" (misclassified as "tripod") can be stealthily perturbed to be correctly classified. Perturbations are rescaled for display.

framework of (non)-robust features (Jetley et al., 2018; Ilyas et al., 2019), many efforts have been devoted to understanding and mitigating the risk raised by adversarial examples, thus closing the gap $\mathcal{G}_{\text{adv}}$. Previous works mainly concern the adversarial risk on correctly classified examples. However, they typically neglect a risk on misclassified examples themselves which will be formalized in this work.

In this paper, we first investigate an intriguing, yet far overlooked phenomenon, where given a DNN classifier $f$ and a *misclassified* example $x$ with class label $y$ (i.e., $f(x) \neq y$), we can easily perturb $x$ to $x_{\text{hyp}}$ such that $f(x_{\text{hyp}}) = y$ and $x_{\text{hyp}} \in \mathcal{B}_\epsilon(x)$. Such an example $x_{\text{hyp}}$ looks harmless, but actually can be maliciously utilized by a *false friend* to fool a model to be self-satisfied. Thus we name them *hypocritical examples* (see Figure 1 for a comparison with adversarial examples).

Adversarial examples and hypocritical examples are two sides of the same coin. On the one side, a well-performed but sensitive model becomes unreliable in the existence of adversaries. On the other side, a poorly performed but sensitive model behaves well with the help of friends. With false friends like these, a naturally trained suboptimal model could have state-of-the-art performance, and even worse, a randomly initialized model could behave like a well-trained one (see Section 2.1).

It is natural then to wonder: *Why should we care about hypocritical examples?* Here we give two main reasons:

1. This is of scientific interest. Hypocritical examples are the opposite of adversarial examples. While adversarial examples are hard test data to a model, hypocritical examples aim to make it easy to do correct classification. Hypocritical examples warn ML researchers to

think carefully about high test accuracy: Does our model truly achieve human-like intelligence, or is it just simply because the test data prefers the model?

2. There are practical threats. A variety of nefarious ends may be achievable if the mistakes of ML systems can be covered up by hypocritical attackers. For instance, before allowing autonomous vehicles to drive on public roads, manufacturers must first pass tests in specific environments (closed or open roads) to obtain a license (Administration et al., 2016; Briefs, 2015; Lei, 2018). An attacker may add imperceptible perturbations on the test examples (e.g., the "stop sign" on the road) stealthily without human notice, to hypocritically help an ML-based autonomous vehicle to pass the tests that might otherwise fail. However, the high performance can not be maintained on public roads without the help of the attacker. Thus, the potential risk is underestimated and traffic accidents might happen unexpectedly when the vehicle driving on public roads.

In such a case, if the examples used to evaluate a model are falsified by a false friend, the model will manifest like a perfect one (on hypocritical examples), but it actually may not be well performed even on clean examples, not to mention adversarial examples. Thus a new imperfection of the classifier can be found in $\mathcal{G}_{\text{hyp}} = \text{Acc}(\mathcal{F}) - \text{Acc}(\mathcal{D})$, the performance gap between the accuracy evaluated on clean set sampled from $\mathcal{D}$ and hypocritically perturbed set $\mathcal{F}$. Still, $\mathcal{F}$ looks no different from $\mathcal{D}$ but can stealthily upgrade the performance. Once a deployer trusts the hypocritical performance carefully designed by a false friend and uses the "well-performed" model in real-world applications, potential security concerns appear even in benign environments. Thus we need methods to defend our models from false friends, that is, making our models have self-knowledge.

We propose a defense method by improving model robustness against hypocritical perturbations. Specifically, we formalize the *hypocritical risk* and minimize it via a differentiable surrogate loss (Section 3). Experimentally, we verify the effectiveness of our proposed attack (Section 2.1) and defense (Section 4.1). Further, we study the transferability of hypocritical examples across models trained with various methods (Section 4.2). Finally, we conclude our paper by discussing and summarizing our results (Section 5 and Section 6). Our main contributions are:

- We give a formal definition of hypocritical examples. We demonstrate the unreliability of standard evaluation process in the existence of false friends and show the potential security risk on the deployment of a model with high hypocritical performance.

- We formalize the hypocritical risk and analyze its relation with natural risk and adversarial risk. We propose the first defense method specialized for hypocritical examples by minimizing the tradeoff between the natural risk and an upper bound of hypocritical risk.

- Extensive experiments verify the effectiveness of our proposed methods. We also examine the transferability of hypocritical examples. We show that the transferability is not always desired by the attackers, which depends on their purpose.

## 2 FALSE FRIENDS AND ADVERSARIES

Better an open enemy than a false friend! Only by being aware of the potential risk of the false friend can we prevent it. In this section, we expose a kind of false friends, who are capable of manipulating model performance stealthily during the evaluation process, thus making the evaluation results unreliable.

We consider a classification task with data $(\boldsymbol{x}, y) \in \mathbb{R}^d \times \{1, \ldots, C\}$ from a distribution $\mathcal{D}$. Denote by $f : \mathbb{R}^d \to \{1, ..., C\}$ the classifier which predicts the class of an input example $\boldsymbol{x}$: $f(\boldsymbol{x}) = \arg\max_k p_k(\boldsymbol{x})$, where $p_k(\boldsymbol{x})$ is the $k$th component of $p(\boldsymbol{x}) : \mathbb{R}^d \to \Delta^C$ (e.g., the output after softmax activation), in which $\Delta^C = \{\boldsymbol{u} \in \mathbb{R}^C \mid \mathbf{1}^T \boldsymbol{u} = 1, \boldsymbol{u} \geq \mathbf{0}\}$ is the probabilistic simplex.

Adversarial examples are *malicious* inputs crafted by an adversary to induce misclassification. We first give the commonly accepted definition of adversarial examples as follows:

**Definition 1** (Adversarial Examples). *Given a classifier $f$ and a correctly classified input $(\boldsymbol{x}, y) \sim \mathcal{D}$ (i.e., $f(\boldsymbol{x}) = y$), an $\epsilon$-bounded adversarial example is an input $\boldsymbol{x}^* \in \mathbb{R}^d$ such that:*

$$f(\boldsymbol{x}^*) \neq y \quad \text{and} \quad \boldsymbol{x}^* \in \mathcal{B}_\epsilon(\boldsymbol{x}).$$

The assumption underlying this definition is that inputs satisfying $\boldsymbol{x}^* \in \mathcal{B}_\epsilon(\boldsymbol{x})$ preserve the label $y$ of the original input $\boldsymbol{x}$. The reason for the existence of adversarial examples is that a model is overly sensitive to non-semantic changes. Next, we formalize a complementary phenomenon to adversarial examples, called hypocritical examples. Hypocritical examples are *malicious* inputs crafted by a false friend to stealthily correct the prediction of a model:

**Definition 2** (Hypocritical Examples). *Given a classifier $f$ and a misclassified input $(\boldsymbol{x}, y) \sim \mathcal{D}$ (i.e., $f(\boldsymbol{x}) \neq y$), an $\epsilon$-bounded hypocritical example is an input $\boldsymbol{x}^* \in \mathbb{R}^d$ such that:*

$$f(\boldsymbol{x}^*) = y \quad and \quad \boldsymbol{x}^* \in \mathcal{B}_\epsilon(\boldsymbol{x}).$$

The same as adversarial examples, hypocritical examples are bounded to preserve the label of the original input, and are another consequence that arises from excessive sensitivity of a classifier.

As a false friend, a hypocritical example can be generated from a misclassified example by maximizing

$$\max_{\boldsymbol{x}' \in \mathcal{B}_\epsilon(\boldsymbol{x})} \mathbb{1}(f(\boldsymbol{x}') = y), \tag{1}$$

which is equivalent to minimizing

$$\min_{\boldsymbol{x}' \in \mathcal{B}_\epsilon(\boldsymbol{x})} \mathbb{1}(f(\boldsymbol{x}') \neq y), \tag{2}$$

where $\mathbb{1}(\cdot)$ is the indicator function. Similar to Madry et al. (2018); Wang et al. (2020b), in practice, we leverage the commonly used cross entropy (CE) loss as the surrogate loss of $\mathbb{1}(f(\boldsymbol{x}') \neq y)$ and minimize it by projected gradient descent (PGD).

Note that Equation 2 looks similar to but conceptually differs from the known targeted adversarial attack (Carlini & Wagner, 2017), which generates a kind of adversarial examples defined on correctly classified clean inputs and targeted to wrong classes. The hypocritical examples here are defined on misclassified inputs and are targeted to their right classes.

## 2.1 ATTACK RESULTS

In this subsection, we demonstrate the power of our proposed hypocritical attack on three benchmark datasets: MNIST (LeCun et al., 1998), CIFAR-10 (Krizhevsky et al., 2009) and ImageNet (Russakovsky et al., 2015).

Table 1: Accuracy (%) evaluated on MNIST. Attacks are bounded with $\epsilon = 0.2$.

| Model | $\mathcal{F}$ | $\mathcal{D}$ | $\mathcal{A}$ |
|---|---|---|---|
| Naive (MLP) | 100.0 | 10.4 | 0.0 |
| Naive (LeNet) | 79.2 | 10.1 | 0.0 |
| Standard (MLP) | 100.0 | 97.8 | 29.8 |
| Standard (LeNet) | 100.0 | 99.4 | 0.1 |

Table 2: Accuracy (%) evaluated on ImageNet. Attacks are bounded with $\epsilon = 16/255$.

| Model | $\mathcal{F}$ | $\mathcal{D}$ | $\mathcal{A}$ |
|---|---|---|---|
| Naive (VGG16) | 100.0 | 0.1 | 0.0 |
| Naive (ResNet50) | 12.6 | 0.1 | 0.0 |
| Standard (VGG16) | 99.9 | 71.6 | 0.3 |
| Standard (ResNet50) | 99.9 | 76.1 | 0.0 |

We attack models trained with standard approach using clean examples (Standard) and models that randomly initialized without training (Naive). For MNIST, the hypocritically perturbed set $\mathcal{F}$ and the adversarially perturbed set $\mathcal{A}$ are constructed by attacking every example in the clean test set sampled from $\mathcal{D}$. Both attacks are bounded by a $l_\infty$ ball with radius $\epsilon = 0.2$. For ImageNet, $\mathcal{F}$ and $\mathcal{A}$ are constructed based on its validation set sampled from $\mathcal{D}$. Both attacks are bounded by a $l_\infty$ ball with radius $\epsilon = 16/255$. For each experiment, we conduct 3 trials with different random seeds and report the averaged result to reduce the impact of random variations. Appendix A.2 describes further experimental details about DNN architecture, training procedure and more results.

Results on MNIST and ImageNet are summarized in Table 1 and Table 2, respectively. First, we find that the naturally trained models are extremely sensitive to hypocritical perturbations (e.g., Standard (MLP) and Standard (LeNet) achieve 100% accuracy on hypocritically perturbed MNIST test set, and Standard (VGG16) and Standard (ResNet50) achieve 99% accuracy on hypocritically perturbed ImageNet validation set). Second, we find that part of randomly initialized models is extremely

sensitive (e.g., Naive (MLP) and Naive (VGG16) achieve $100\%$ accuracy on $\mathcal{F}$ of MNIST and ImageNet). These results demonstrate the unreliability of standard evaluation process in the existence of false friends. Once a "well-performed" model (such as Naive (MLP) or Naive (VGG16)) is permitted to be deployed in real-world applications due to that the deployer has a false sense of performance, potential security concerns appear even in benign environments.

It seems that the Naive (ResNet50) model is relatively robust to hypocritical examples on ImageNet. But that is just a trivial defense, it simply predicts most of the points in input region as a certain class because of the poor scaling of network weights at initialization (He et al., 2016b; Elsayed et al., 2019). More discussions are in Appendix A.2. Therefore, it is not enough to blindly pursue robustness against hypocritical perturbations but ignore the performance on clean examples.

## 3 HYPOCRITICAL RISK

In this section, we formalize the hypocritical risk and analyze the relation between natural risk, adversarial risk, and hypocritical risk. We propose a defense method specialized for hypocritical examples by minimizing the tradeoff between natural risk and an upper bound of the hypocritical risk. Moreover, by decomposing a existing method designed for adversarial defense (TRADES (Zhang et al., 2019)), we find that, surprisingly, TRADES minimizes not only the adversarial risk on correctly classified examples, but also a looser upper bound of the hypocritical risk. Our theoretical analysis suggests that TRADES can be another candidate defense method for hypocritical examples.

To characterize the adversarial robustness of a classifier $f$, Madry et al. (2018); Uesato et al. (2018); Cullina et al. (2018) defined the *adversarial risk* under the threat model of bounded $\epsilon$ ball:

$$\mathcal{R}_{\mathsf{adv}}(f) = \mathop{\mathbb{E}}_{(\boldsymbol{x},y)\sim\mathcal{D}}\left[\max_{\boldsymbol{x}'\in\mathcal{B}_\epsilon(\boldsymbol{x})} \mathbb{1}(f(\boldsymbol{x}') \neq y)\right]. \tag{3}$$

The standard measure of classifier performance, known as *natural risk*, is denoted as $\mathcal{R}_{\mathsf{nat}}(f) = \mathbb{E}_{(\boldsymbol{x},y)\sim\mathcal{D}}[\mathbb{1}(f(\boldsymbol{x}) \neq y)]$. Let $q(\boldsymbol{x},y)$ be the probability density function of data distribution $\mathcal{D}$. We denote by $\mathcal{S}_f^+$ the conditional data distribution on correctly classified examples w.r.t. $f$, with a conditional density function $q(\boldsymbol{x},y \mid E) = q(\boldsymbol{x},y)/\mathrm{Z}(E)$ if $E$ is true (otherwise $q(\boldsymbol{x},y \mid E) = 0$), where the event $E$ is $f(\boldsymbol{x}) = y$ and $\mathrm{Z}(E) = \int_{\boldsymbol{x},y} \mathbb{1}(f(\boldsymbol{x}) = y)dq(\boldsymbol{x},y)$ is a normalizing constant. We denote by $\mathcal{S}_f^-$ the conditional data distribution on misclassified examples with the conditional density function $q(\boldsymbol{x},y \mid E)$ and $f(\boldsymbol{x}) \neq y$ as the event $E$. Then we have the following relation between the natural risk and the adversarial risk:

**Proposition 1.** *Denote the adversarial risk on correctly classified examples by*

$$\hat{\mathcal{R}}_{\mathsf{adv}}(f) = \mathop{\mathbb{E}}_{(\boldsymbol{x},y)\sim\mathcal{S}_f^+}\left[\max_{\boldsymbol{x}'\in\mathcal{B}_\epsilon(\boldsymbol{x})} \mathbb{1}(f(\boldsymbol{x}') \neq y)\right],$$

*then we have*

$$\mathcal{R}_{\mathsf{adv}}(f) = \mathcal{R}_{\mathsf{nat}}(f) + (1 - \mathcal{R}_{\mathsf{nat}}(f))\hat{\mathcal{R}}_{\mathsf{adv}}(f).$$

Proposition 1 shows that we can view the adversarial risk $\mathcal{R}_{\mathsf{adv}}(f)$ as the tradeoff between $\mathcal{R}_{\mathsf{nat}}(f)$ and $\hat{\mathcal{R}}_{\mathsf{adv}}(f)$ with the scaling parameter $\lambda = 1 - \mathcal{R}_{\mathsf{nat}}(f)$. The adversarial risk on correctly classified examples $\hat{\mathcal{R}}_{\mathsf{adv}}(f)$ is in sharp contrast to the hypocritical risk defined on misclassified examples formalized as follows:

**Definition 3** (Hypocritical Risk). *The hypocritical risk on misclassified examples of a classifier $f$ under the threat model of bounded $\epsilon$ ball is defined as*

$$\hat{\mathcal{R}}_{\mathsf{hyp}}(f) = \mathop{\mathbb{E}}_{(\boldsymbol{x},y)\sim\mathcal{S}_f^-}\left[\max_{\boldsymbol{x}'\in\mathcal{B}_\epsilon(\boldsymbol{x})} \mathbb{1}(f(\boldsymbol{x}') = y)\right].$$

The hypocritical risk $\hat{\mathcal{R}}_{\mathsf{hyp}}(f)$ is the proportion of perturbed examples (originally misclassified) that can be successfully correctly classified by the classifier after a false friend's attack. When considering the existence of false friends, a good model should have not only low natural risk but also low hypocritical risk, to be robust against hypocritical perturbations.

## 3.1 TRADEOFF BETWEEN NATURAL AND HYPOCRITICAL RISKS

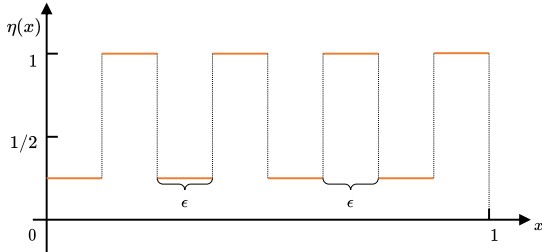

Figure 2: Counterexample given by Equation 4.

Motivated by the tradeoff between natural and adversarial risks Tsipras et al. (2019); Zhang et al. (2019), we notice that there may also exist an inherent tension between the goal of natural risk minimization and hypocritical risk minimization. To illustrate the phenomenon, we provide a toy example here, which is modified from the example in Zhang et al. (2019), and its risk minimization solutions can be analytically found.

Consider the case $(x, y) \in \mathbb{R} \times \{-1, +1\}$ from a distribution $\mathcal{D}$, where the marginal distribution over the instance space is a uniform distribution over $[0, 1]$, and for $k = 0, 1, \cdots, \lceil \frac{1}{2\epsilon} - 1 \rceil$,

$$
\begin{aligned}
\eta(x) &:= \Pr(y = +1 \mid x) \\
&= \begin{cases} 1/4, & x \in [2k\epsilon, (2k+1)\epsilon), \\ 1, & x \in ((2k+1)\epsilon, (2k+2)\epsilon]. \end{cases}
\end{aligned}
\tag{4}
$$

See Figure 2 for visualization of $\eta(x)$. In this problem, we consider two classifiers: a) the Bayes optimal classifier $\text{sign}(2\eta(x) - 1)$; b) the all-one classifier which always outputs "positive". Table 3 displays the trade-off between natural and hypocritical risks: the minimal natural risk $1/8$ is achieved by the Bayes optimal classifier with large hypocritical risk, while the optimal hypocritical risk $0$ is achieved by the all-one classifier with large natural risk.

Table 3: Comparison of Bayes optimal classifier and all-one classifier.

|  | Bayes Optimal Classifier | All-One Classifier |
|---|---|---|
| $\mathcal{R}_{\text{nat}}$ | 1/8 (optimal) | 3/8 |
| $\hat{\mathcal{R}}_{\text{hyp}}$ | 1 | 0 (optimal) |

## 3.2 UPPER BOUNDS OF HYPOCRITICAL RISK

It is natural then to optimize our models to minimize natural and hypocritical risks at the same time. However, it's hard to do optimization over $\hat{\mathcal{R}}_{\text{hyp}}(f)$. To ease the optimization obstacles in there, we derive the following upper bounds.

**Theorem 1.** *For any data distribution $\mathcal{D}$ and its corresponding conditional distribution on misclassified examples $\mathcal{S}_f^-$ w.r.t. a classifier $f$, we have*

$$
\underbrace{\mathbb{E}_{(\boldsymbol{x},y)\sim\mathcal{S}_f^-} \mathbb{1}(f(\boldsymbol{x}_{\text{hyp}}) = y)}_{\hat{\mathcal{R}}_{\text{hyp}}(f)} \leq \underbrace{\mathbb{E}_{(\boldsymbol{x},y)\sim\mathcal{S}_f^-} \mathbb{1}(f(\boldsymbol{x}_{\text{hyp}}) \neq f(\boldsymbol{x}))}_{\overline{\mathcal{R}}_{\text{hyp}}(f)} \leq \underbrace{\mathbb{E}_{(\boldsymbol{x},y)\sim\mathcal{S}_f^-} \mathbb{1}(f(\boldsymbol{x}_{\text{rev}}) \neq f(\boldsymbol{x}))}_{\overline{\overline{\mathcal{R}}}_{\text{hyp}}(f)},
$$

*where $\boldsymbol{x}_{\text{hyp}} = \arg\max_{\boldsymbol{x}' \in \mathcal{B}_\epsilon(\boldsymbol{x})} \mathbb{1}(f(\boldsymbol{x}') = y)$ and $\boldsymbol{x}_{\text{rev}} = \arg\max_{\boldsymbol{x}' \in \mathcal{B}_\epsilon(\boldsymbol{x})} \mathbb{1}(f(\boldsymbol{x}') \neq f(\boldsymbol{x}))$.*

Here $\boldsymbol{x}_{\text{rev}}$ means that it pursues to reverse a clean example to a different class, from the point of view of the model. The upper bounds found in Theorem 1 allow us to optimize the hypocritical risk using

proper surrogate loss functions which are both physical meaningful and computaionally tractable. Before moving forward to algorithmic design, we state a useful proposition below, which reveals the internal mechanism behind in TRADES.

**Proposition 2.** $\mathcal{R}_{\text{rev}}(f) = (1 - \mathcal{R}_{\text{nat}}(f))\hat{\mathcal{R}}_{\text{adv}}(f) + \mathcal{R}_{\text{nat}}(f)\overline{\overline{\mathcal{R}}}_{\text{hyp}}(f) = \mathbb{E}_{(\boldsymbol{x},y)\sim\mathcal{D}} \mathbb{1}(f(\boldsymbol{x}_{\text{rev}}) \neq f(\boldsymbol{x})).$

Proposition 2 shows a connection between adversarial risk and hypocritical risk: the adversarial risk on correctly classified examples $\hat{\mathcal{R}}_{\text{adv}}(f)$ and the looser upper bound of the hypocritical risk on misclassified examples $\overline{\overline{\mathcal{R}}}_{\text{hyp}}(f)$ can be seamlessly united to a new risk on all examples $\mathcal{R}_{\text{rev}}(f)$. We name it *reversible risk* since minimizing it pursues the model whose predictions can't be reversed by small perturbations.

### 3.3 ALGORITHMIC DESIGN

Now we are ready to design objective functions that improve model robustness against hypocritical examples while keeping model accuracy on clean examples.

Similar to Zhang et al. (2019); Wang et al. (2020b); Tramèr et al. (2020); Raghunathan et al. (2020), we propose a defense objective by minimizing the tradeoff between the natural risk and the tighter upper bound of the hypocritical risk:

$$\mathcal{R}_{\text{THRM}}(f) = \mathcal{R}_{\text{nat}}(f) + \lambda\overline{\mathcal{R}}_{\text{hyp}}(f), \tag{5}$$

where $\lambda > 0$ is a tunable scaling parameter balancing the importance of natural risk and hypocritical risk. We name our method **THRM** (Tradeoff for Hypocritical Risk Minimization).

Optimization over 0-1 loss in THRM is still intractable. In practice, for the indicator function $\mathbb{1}(f(\boldsymbol{x}) \neq y)$ in $\mathcal{R}_{\text{nat}}(f)$, we adopt the commonly used CE loss as surrogate loss. Observed that $\overline{\mathcal{R}}_{\text{hyp}}(f) = \frac{1}{\mathcal{R}_{\text{nat}}(f)}\mathbb{E}_{(\boldsymbol{x},y)\sim\mathcal{D}}\mathbb{1}(f(\boldsymbol{x}_{\text{hyp}}) \neq f(\boldsymbol{x}))$, we absorb the $\mathcal{R}_{\text{nat}}(f)$ term into $\lambda$ and use KL divergence as the surrogate loss of the indicator function $\mathbb{1}(f(\boldsymbol{x}_{\text{hyp}}) \neq f(\boldsymbol{x}))$ (Zheng et al., 2016; Zhang et al., 2019; Wang et al., 2020b), since $f(\boldsymbol{x}_{\text{hyp}}) \neq f(\boldsymbol{x})$ implies that the perturbed examples have different output distributions to that of clean examples. Our final objective function for THRM becomes

$$\mathcal{L}_{\text{THRM}} = \mathbb{E}_{(\boldsymbol{x},y)\sim\mathcal{D}} \left[\mathcal{L}_{\text{CE}}(p(\boldsymbol{x}), y) + \lambda\mathcal{L}_{\text{KL}}(p(\boldsymbol{x}), p(\boldsymbol{x}_{\text{hyp}}))\right]. \tag{6}$$

Intuition behind the objective $\mathcal{L}_{\text{THRM}}$: the first term in Equation 6 encourages the natural risk to be optimized, while the second regularization term encourages the output to be stable against hypocritical perturbations, that is, the classifier should not be overly confident in its predictions especially when a false friend wants it to be.

To derive the objective function for TRADES, we can minimize the tradeoff between the natural risk and the reversible risk:

$$\mathcal{R}_{\text{TRADES}}(f) = \mathcal{R}_{\text{nat}}(f) + \lambda\mathcal{R}_{\text{rev}}(f). \tag{7}$$

Similar to THRM, we use CE loss and KL divergence as the surrogate loss of $\mathbb{1}(f(\boldsymbol{x}) \neq y)$ and $\mathbb{1}(f(\boldsymbol{x}_{\text{rev}}) \neq f(\boldsymbol{x}))$, respectively. The final objective function becomes

$$\mathcal{L}_{\text{TRADES}} = \mathbb{E}_{(\boldsymbol{x},y)\sim\mathcal{D}} \left[\mathcal{L}_{\text{CE}}(p(\boldsymbol{x}), y) + \lambda\mathcal{L}_{\text{KL}}(p(\boldsymbol{x}), p(\boldsymbol{x}_{\text{rev}}))\right], \tag{8}$$

which is exactly the multi-class classification objective function first proposed in Zhang et al. (2019) for adversarial defense. From the perspective of the hypocritical risk, our Proposition 2 reveals an advantage behind it, that is, TRADES is capable of minimizing the upper bound of hypocritical risk $\overline{\overline{\mathcal{R}}}_{\text{hyp}}(f)$, thus can be considered as a candidate defense method for hypocritical examples. Proposition 2 also implies that there may be a deeper connection between adversarial robustness and hypocritical robustness. We will discuss it more and compare our proposed THRM with TRADES in next section.

## 4 EXPERIMENTS

In this section, to verify the effectiveness of the methods (THRM and TRADES) suggested in Section 3.3, we conduct experiments on real-world datasets including MNIST and CIFAR-10.

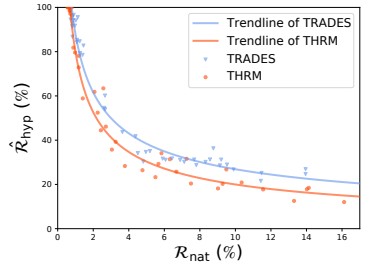 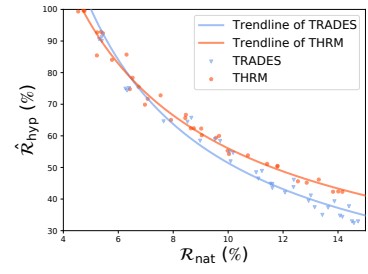

(a) On MNIST. Perturbations are bounded by $l_\infty$ norm with $\epsilon = 0.2$.

(b) On CIFAR-10. Perturbations are bounded by $l_\infty$ norm with $\epsilon = 2/255$.

Figure 3: Tradeoff between natural risk and hypocritical risk on real-world datasets.

### 4.1 WHITE-BOX ANALYSIS

For the wide range of the scaling parameter $\lambda$, we conduct experiments in parallel over multiple NVIDIA Tesla V100 GPUs. On MNIST, perturbations are bounded by $l_\infty$ norm with $\epsilon = 0.2$. On CIFAR-10, models are trained against 3 different hypocritical attackers bounded by $l_\infty$ norm with $\epsilon = 1/255$, $\epsilon = 2/255$ and $\epsilon = 8/255$, respectively. Each experiment is conducted 3 times with different random seeds. The hypocritical risk reported here is actually an approximation of the real value, since the optimization problem in it is NP-hard and we approximately solve it using surrogate loss and PGD on test set. Further details about model architecture and training procedure are in Appendix A.3. Note that these experiments are extensive. It takes over 230 GPU days to completely train the models considered in this section. We believe that these experiments are beneficial to the ML community to further understand the tradeoffs and relative merits in THRM and TRADES.

Results on MNIST ($\epsilon = 0.2$) and CIFAR10 ($\epsilon = 2/255$) are shown in Figure 3. Each data point represents a model trained with different $\lambda$. More results including comparison with Madry's defense (Madry et al., 2018) are provided in Appendix A.3 due to the limited space. First, we observe that, on both datasets, as the regularization parameter $\lambda$ increases, the natural risk $\mathcal{R}_{\text{nat}}$ increases while the hypocritical risk $\hat{\mathcal{R}}_{\text{hyp}}$ decreases, which verifies the effectiveness of our proposed method and the theoretical analysis in Proposition 2, where we reveal that TRADES is capable of minimizing a looser upper bound of hypocritical risk. Second, we show that THRM achieves better tradeoff on MNIST since it optimizes a tighter upper bound than TRADES. However, the situation becomes nuanced on CIFAR-10. As we can see in Figure 3(b), THRM seems to behave better in the beginning when $\lambda$ is small but is surpassed by TRADES when $\lambda$ increases. Overall, optimizing only a tighter upper bound of hypocritical risk achieves better tradeoff on test set when the task is relatively simple (e.g., on MNIST with $\epsilon = 0.2$), while simultaneously optimizing hypocritical risk and adversarial risk achieves better tradeoff on test set when the task tends to be hard (e.g., on CIFAR-10 with $\epsilon = 2/255$ and $\epsilon = 8/255$).

Above phenomenon shows that, when dealing with finite sample size and finite-time gradient-descent trained classifiers, better adversarial robustness may help the generalization of hypocritical robustness, which conforms our intuition that they are two sides of the same coin. Interestingly, a contemporary work claims that, on CIFAR-10, TRADES achieves better adversarial robustness than Madry's defense in fair hyperparameter settings (Anonymous, 2021). Thus there may be potential mutual benefits between adversarial robustness and hypocritical robustness. After all, robust training objectives force DNNs to be invariant to signals that humans are invariant to, which may lead to feature representations that are more similar to what humans use (Salman et al., 2020). A rigorous treatment of the synergism is beyond the scope of the current paper but is an important future direction.

### 4.2 TRANSFERABILITY ANALYSIS

Transferability of adversarial examples across models is well known (Tramèr et al., 2017; Papernot et al., 2017b; Ilyas et al., 2019) and here we examine the transferability of hypocritical examples

on MNIST and CIFAR-10. We observe that hypocritical examples, *i)* can transfer easily between naturally trained models, *ii)* are hard to transfer from randomly initialized models to other models (and vise versa), *iii)* are hard to transfer from standard models to defended models, *iv)* generated from THRM models usually have high transferability. Experimental details are in Appendix A.4. Better transferability is beneficial for black-box attacks but is not always desired by hypocritical attackers. A hypocritical attacker only expects high transferability on the targeted model the attacker chose to help. If there are other competing models available to the deployer, the attacker actually does not want the hypocritical examples to be successfully transferred to those competing models. Thus fine-grained attack methods are required. We leave this to future work.

## 5 DISCUSSION

The false friends considered in this paper are as powerful as typical adversaries. They all know the ground truth labels of clean examples. Such powerful friends actually can help a model to not only correctly classify a misclassified clean example but also correctly classify an adversarial example crafted by an adversary. One may expect to rely on true friends against adversaries. Unfortunately, an omniscient and faithful friend is unachievable in practical tasks, so far at least. Once it is achieved, the problem of robustness disappears immediately. What we can do at present is using a relatively more robust model as a surrogate of the true friend to improve the robustness of a weak model. This induces a promising general method in practice, that is, high-performance models can be employed as true friends to help a weak model without exposing training data and model weights for the purpose of privacy protection and knowledge transfer (Abadi et al., 2016; Papernot et al., 2017a). Additional discussions are in Appendix C.

## 6 CONCLUSION

In this work, we expose a new risk arising from excessive sensitivity. Model performance becomes hypocritical in the existence of false friends. By formalizing the hypocritical risk and analyzing its relation with natural risk and adversarial risk, we propose to use THRM and TRADES as defense methods against hypocritical perturbations. Extensive experiments verify the effectiveness of methods. These findings open new avenues for mitigating and exploiting model sensitivity.

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

# A  EXPERIMENTAL DETAILS

## A.1  DETAILS IN FIGURE 1

**Attack procedure.** In adversarial attacks, we perturb clean inputs to maximize the surrogate loss using PGD. In hypocritical attacks, we perturb clean inputs to minimize the surrogate loss using PGD. In both attacks, for the purpose of imperceptibility, we execute PGD attack 100 steps (step size is $\epsilon/50$) with early stopping on ImageNet and the budget $\epsilon$ here is $2/255$.

**More examples.** More adversarial examples and hypocritical examples generated on ImageNet using our methods are shown in Figure 5. More hypocritical examples generated on MNIST and CIFAR-10 are shown in Figure 4(a) and Figure 4(b). The victim models are LeNet (Standard) and Wide ResNet (Standard) for MNIST and CIFAR-10, respectively. They are trained with the same procedures described in Appendix A.2. In both attacks, for the purpose of imperceptibility, we execute 100 steps PGD attacks (step size is $\epsilon/50$) with early stopping on MNIST and CIFAR-10. The budget $\epsilon$ for MNIST here is $0.2$. The budget for CIFAR-10 here is $8/255$.

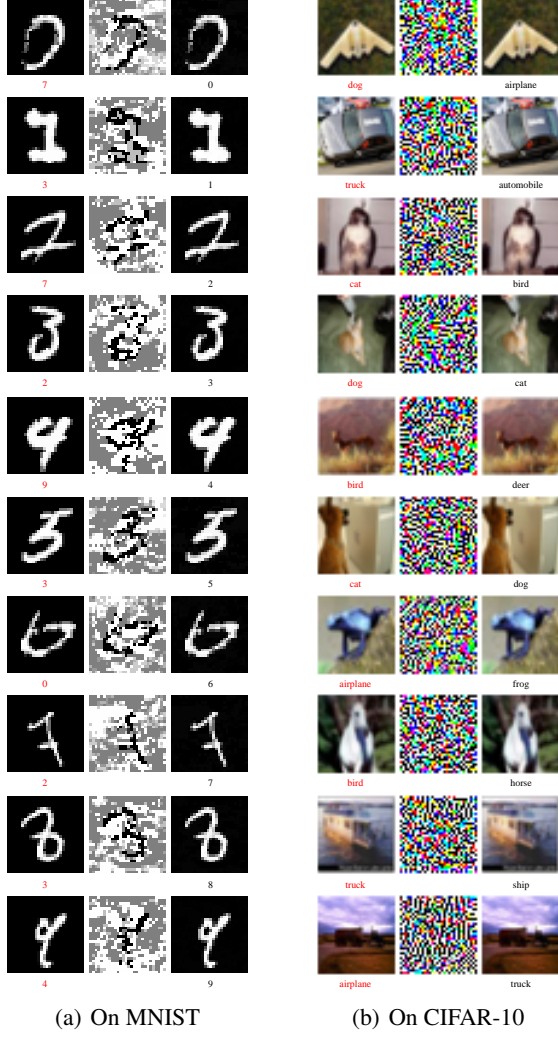

(a) On MNIST                    (b) On CIFAR-10

Figure 4: Hypocritical examples. In each subfigure, the first cloumn represents the clean examples sampled from original data distribution, the second cloumn represents the generated perturbations, the third cloumn represents the perturbed examples. Perturbations are rescaled for display. Red labels and black labels below images denote misclassification and correct classification, respectively.

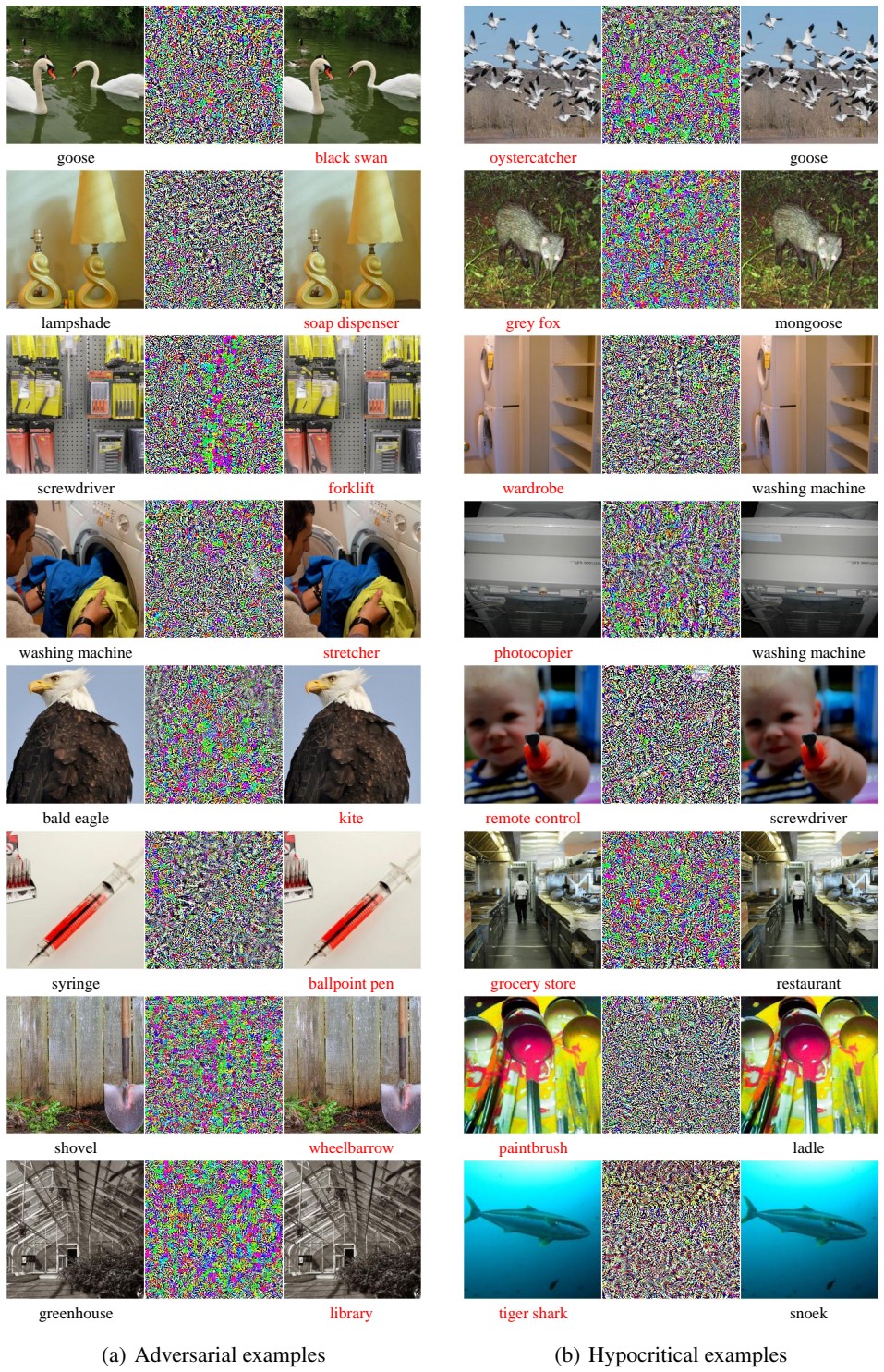

(a) Adversarial examples          (b) Hypocritical examples

Figure 5: More examples on ImageNet. In each subfigure, the first cloumn represents the clean examples sampled from original data distribution, the second cloumn represents the generated perturbations, the third cloumn represents the perturbed examples. Perturbations are rescaled for display. The model predictions of these images are shown below each image. Red labels and black labels below images denote misclassification and correct classification, respectively.

## A.2 Details in Section 2.1

**Architecture.** For MNIST, a four-layer multilayer perception (MLP) (2 hidden layers, 768 neurons in each) with ReLU activations and a variant of LeNet model (2 convolutional layers of sizes 32 and 64, and a fully connected layer of size 1024) are adopted. For CIFAR-10, a four-layer MLP (2 hidden layers, 3072 neurons in each) with ReLU activations, a ResNet18 (He et al., 2016a) and a Wide ResNet (Zagoruyko & Komodakis, 2016) (with depth 28 and width factor 10) are adopted. For ImageNet, a VGG16 (Simonyan & Zisserman, 2014) and a ResNet50 (He et al., 2016a) are adopted.

**Training procedure.** *i)* Models trained with standard approach using clean examples (Standard). For MNIST, models are trained for 80 epochs with Adam optimizer with batch size 128 and a learning rate of 0.001. Early stopping is done with holding out 1000 examples from the MNIST training set. For CIFAR-10, models are trained for 150 epochs with SGD optimizer with batch size 128 and the learning rate starts with 0.1, and is divided it by 10 at 90 and 125 epochs. We apply weight decay of 2e-4 and momentum of 0.9. Early stopping is done with holding out 1000 examples from the CIFAR-10 training set. For ImageNet, we use the pretrained standard models available within PyTorch (torchvision.models). *ii)* Models that randomly initialized without training (Naive). For all models, we use the default PyTorch initialization, except that we initialize the convolutional weights in Wide ResNet with He initialization (He et al., 2015). We conduct all the experiments using a single NVIDIA Tesla V100 GPU. Each experiment is conducted 3 times with different random seeds, except the standard models trained on ImageNet, in which we use the pretrained standard models available within PyTorch.

**Attack procedure.** In adversarial attacks, we perturb clean inputs to maximize the surrogate loss using PGD. In hypocritical attacks, we perturb clean inputs to minimize the surrogate loss using PGD. In both attacks, we execute 50 steps PGD attacks (step size is $\epsilon/10$) with 20 times of random restart on MNIST and CIFAR-10, and we use 50 steps PGD attacks (step size is $\epsilon/8$) on ImageNet. Other hyperparameter choices didn't offer a significant change in accuracy. On MNIST, the hypocritical perturbed set $\mathcal{F}$ and the adversarially perturbed set $\mathcal{A}$ are constructed by attacking every example in the clean test set sampled from $\mathcal{D}$. Both attacks are bounded by a $l_\infty$ ball with radius $\epsilon = 0.2$. On CIFAR-10, both attacks are bounded by a $l_\infty$ ball with radius $\epsilon = 8/255$. On ImageNet, $\mathcal{F}$ and $\mathcal{A}$ are constructed based on its validation set sampled from $\mathcal{D}$. Both attacks are bounded by a $l_\infty$ ball with radius $\epsilon = 16/255$.

**Numerical results.** The attack results on CIFAR-10 are shown in Table 4. Full results of Table 1, Table 2 and Table 4 are shown in Table 5, Table 6 and Table 7, respectively. Moreover, we show the attack results of 9 Naive models evaluated on ImageNet in Table 6. We find that all the Naive models in VGG family achieve high accuracy on $\mathcal{F}$ and all the Naive models in ResNet family have relatively poor performance on $\mathcal{F}$. Especially, the Naive (ResNet152) model in Trial 1 is invariant to hypocritical perturbations. Even in the existence of a strong false friend, the hypocritical performance is still as low as the clean performance (only $0.1\%$). We carefully examined the Naive (ResNet152) model and find that it's actually a trivial classifier, which purely classifies almost all the points in input region $[0,1]^d$ as a certain class for some simple reasons, such as poor scaling of network weights at initialization. Therefore, it is not enough to blindly pursue robustness against hypocritical perturbations but ignore the performance on clean examples. Once we train a Naive model with clean examples, the model becomes vulnerable immediately (see Standard (ResNet50)), whereas the trained weights are better conditioned (Elsayed et al., 2019).

Table 4: Accuracy (%) of models evaluated on CIFAR-10. Attacks are bounded with $\epsilon = 8/255$.

| Model | $\mathcal{F}$ | $\mathcal{D}$ | $\mathcal{A}$ |
|---|---|---|---|
| Naive (MLP) | 92.3 | 9.9 | 0.0 |
| Naive (ResNet18) | 20.8 | 8.7 | 0.3 |
| Naive (Wide ResNet) | 13.8 | 10.0 | 6.9 |
| Standard (MLP) | 88.6 | 45.1 | 3.9 |
| Standard (ResNet18) | 100.0 | 94.1 | 0.0 |
| Standard (Wide ResNet) | 100.0 | 95.1 | 0.0 |

Table 5: Full results of accuracy (%) evaluated on MNIST. Attacks are bounded with $\epsilon = 0.2$.

| Model | Trial 1 | | | Trial 2 | | | Trial 3 | | |
|---|---|---|---|---|---|---|---|---|---|
| | $\mathcal{F}$ | $\mathcal{D}$ | $\mathcal{A}$ | $\mathcal{F}$ | $\mathcal{D}$ | $\mathcal{A}$ | $\mathcal{F}$ | $\mathcal{D}$ | $\mathcal{A}$ |
| Naive (MLP) | 100.0 | 11.2 | 0.0 | 100.0 | 10.8 | 0.0 | 100.0 | 9.4 | 0.0 |
| Naive (LeNet) | 69.1 | 9.7 | 0.0 | 77.5 | 10.6 | 0.0 | 91.0 | 10.1 | 0.0 |
| Standard (MLP) | 100.0 | 98.0 | 31.1 | 100.0 | 97.7 | 30.4 | 100.0 | 97.7 | 28.1 |
| Standard (LeNet) | 100.0 | 99.4 | 0.1 | 100.0 | 99.4 | 0.1 | 100.0 | 99.3 | 0.0 |

Table 6: Full results of accuracy (%) evaluated on ImageNet. Attacks are bounded with $\epsilon = 16/255$.

| Model | Trial 1 | | | Trial 2 | | | Trial 3 | | |
|---|---|---|---|---|---|---|---|---|---|
| | $\mathcal{F}$ | $\mathcal{D}$ | $\mathcal{A}$ | $\mathcal{F}$ | $\mathcal{D}$ | $\mathcal{A}$ | $\mathcal{F}$ | $\mathcal{D}$ | $\mathcal{A}$ |
| Naive (VGG11) | 100.0 | 0.1 | 0.0 | 100.0 | 0.1 | 0.0 | 100.0 | 0.1 | 0.0 |
| Naive (VGG13) | 100.0 | 0.1 | 0.0 | 100.0 | 0.1 | 0.0 | 100.0 | 0.1 | 0.0 |
| Naive (VGG16) | 100.0 | 0.1 | 0.0 | 100.0 | 0.1 | 0.0 | 100.0 | 0.1 | 0.0 |
| Naive (VGG19) | 100.0 | 0.1 | 0.0 | 100.0 | 0.1 | 0.0 | 100.0 | 0.1 | 0.0 |
| Naive (ResNet18) | 58.4 | 0.1 | 0.0 | 83.2 | 0.1 | 0.0 | 57.6 | 0.1 | 0.0 |
| Naive (ResNet34) | 7.4 | 0.1 | 0.0 | 12.5 | 0.1 | 0.0 | 10.4 | 0.1 | 0.0 |
| Naive (ResNet50) | 10.6 | 0.1 | 0.0 | 14.7 | 0.1 | 0.0 | 12.6 | 0.1 | 0.0 |
| Naive (ResNet101) | 0.3 | 0.1 | 0.1 | 0.2 | 0.1 | 0.1 | 0.3 | 0.1 | 0.1 |
| Naive (ResNet152) | 0.1 | 0.1 | 0.1 | 0.3 | 0.1 | 0.1 | 0.2 | 0.1 | 0.1 |
| Standard (VGG16) | 99.9 | 71.6 | 0.3 | N/A | N/A | N/A | N/A | N/A | N/A |
| Standard (ResNet50) | 99.9 | 76.1 | 0.0 | N/A | N/A | N/A | N/A | N/A | N/A |

## A.3 DETAILS IN SECTION 4.1

**Architecture.** For MNIST, a variant of LeNet model (2 convolutional layers of sizes 32 and 64, and a fully connected layer of size 1024) is adopted. For CIFAR-10, a Wide ResNet (with depth 28 and width factor 10) is adopted.

**Training procedure.** For the wide range of the scaling parameter $\lambda$, we conduct experiments in parallel over multiple NVIDIA Tesla V100 GPUs. Each experiment is conducted 3 times with different random seeds. For MNIST, all models (including Standard, Madry, TRADES, THRM) are trained for 80 epochs with Adam optimizer with batch size 128 and a learning rate of 0.001. Early stopping is done with holding out 1000 examples from the MNIST training set as suggested in Rice et al. (2020). For CIFAR-10, all models are trained for 150 epochs with SGD optimizer with batch size 128 and the learning rate starts with 0.1, and is divided it by 10 at 90 and 125 epochs. We apply weight decay of 2e-4 and momentum of 0.9. Early stopping is done with holding out 1000 examples from the CIFAR-10 training set as suggested in Rice et al. (2020).

**Attack procedure.** For the inner maximization in the objective function of THRM, we perturb clean inputs to minimize the CE loss as the surrogate loss. For the inner maximization in TRADES, we maximize the KL divergence as the surrogate loss. For the inner maximization in Madry, we maximize the CE loss as the surrogate loss. On MNIST, the training attack is PGD with random start and 10 iterations (step size $\epsilon/4$). On CIFAR-10, the training attack is PGD with random start and 10 iterations (step size $\epsilon/4$) when $\epsilon = 8/255$, and the training attack is PGD with random start and 7 iterations (step size $\epsilon/3$) when $\epsilon = 1/255$ and $\epsilon = 2/255$. On all experiments, the test attack is 50 steps PGD (step size is $\epsilon/10$) with 20 times of random restart. Other hyperparameter choices didn't offer a significant change in accuracy.

**Numerical results.** The natural risk reported here is estimated on test set. The hypocritical risk reported here is estimated on test set and is actually an approximation of the real value since we approximately solve the optimization problem by PGD on examples from test set. Results on MNIST ($\epsilon = 0.2$) and CIFAR-10 ($\epsilon = 1/255$, $\epsilon = 2/255$ and $\epsilon = 8/255$) are shown in Figure

Table 7: Full results of accuracy (%) evaluated on CIFAR-10. Attacks are bounded with $\epsilon = 8/255$.

| Model | Trial 1 | | | Trial 2 | | | Trial 3 | | |
|---|---|---|---|---|---|---|---|---|---|
| | $\mathcal{F}$ | $\mathcal{D}$ | $\mathcal{A}$ | $\mathcal{F}$ | $\mathcal{D}$ | $\mathcal{A}$ | $\mathcal{F}$ | $\mathcal{D}$ | $\mathcal{A}$ |
| Naive (MLP) | 98.2 | 9.0 | 0.0 | 91.4 | 10.3 | 0.0 | 87.4 | 10.3 | 0.0 |
| Naive (ResNet18) | 14.8 | 10.0 | 0.6 | 20.7 | 9.2 | 0.2 | 27.0 | 6.9 | 0.0 |
| Naive (Wide ResNet) | 12.2 | 10.0 | 7.7 | 10.4 | 10.0 | 9.6 | 18.7 | 10.0 | 3.3 |
| Standard (MLP) | 89.1 | 46.2 | 3.6 | 88.4 | 44.8 | 3.8 | 88.3 | 44.5 | 4.2 |
| Standard (ResNet18) | 100.0 | 93.9 | 0.0 | 100.0 | 94.3 | 0.0 | 100.0 | 94.0 | 0.0 |
| Standard (Wide ResNet) | 100.0 | 95.0 | 0.0 | 100.0 | 95.1 | 0.0 | 100.0 | 95.3 | 0.0 |

6. Each point in Figure 6 represents one model trained with a certain $\lambda$. Full numerical results on MNIST ($\epsilon = 0.2$) and CIFAR-10 ($\epsilon = 1/255$, $\epsilon = 2/255$ and $\epsilon = 8/255$) can be found in Table 8, Table 9, Table 10 and Table 11, respectively. On MNIST ($\epsilon = 0.2$), THRM has better tradeoff than TRADES. However, when the task becomes hard, TRADES performs as well as or better than THRM. On CIFAR-10, as the task becomes harder (the larger the radius $\epsilon$ the harder the task), the gap between TRADES and THRM becomes larger. This phenomenon shows that better adversarial robustness may help the generalization of hypocritical robustness, especially when the task is hard. Moreover, we compare our methods with Madry et al. (2018)'s defense designed for adversarial robustness (denoted as Madry) [1] and standard training method (denoted as Standard). We summarize results in Table 12. For direct comparison, we pick a certain $\lambda$ for each model trained by TRADES and THRM in each task. We observed that, in all tasks, Madry's defense has nonnegligible robustness on hypocritical examples, although there is no hypocritical risk or its upper bound in the objective function. This phenomenon indicates that optimizing only adversarial risk could bring a certain degree of robustness against hypocritical examples. While this experimental results partly support our hypothesis (i.e., the potential mutual benefits between robustness against adversarial perturbations and hypocritical perturbations), we do not take the evidence as an ultimate one and further exploration is needed. We note that the standard deviation becomes larger when $\lambda$ is bigger in TRADES and THRM, which is attributed to optimization difficulty and result in more significant difference among different trials. Reducing the initial learning rate may mitigate this phenomenon.

For completeness, we further evaluate the adversarial risk on correctly classified examples of the models trained by THRM and TRADES. Results on MNIST ($\epsilon = 0.2$) and CIFAR-10 ($\epsilon = 2/255$) are summarized in Table 13 and Table 14, respectively. One interesting finding is that models trained with THRM manifest noteworthy adversarial robustness, especially on CIFAR-10, although there is no adversarial risk in the objective function of THRM. These facts also support the hypothesis (i.e., the potential mutual benefits between robustness against adversarial perturbations and hypocritical perturbations).

---

[1]They actually optimize the adversarial risk in Equation 3 via surrogate loss.

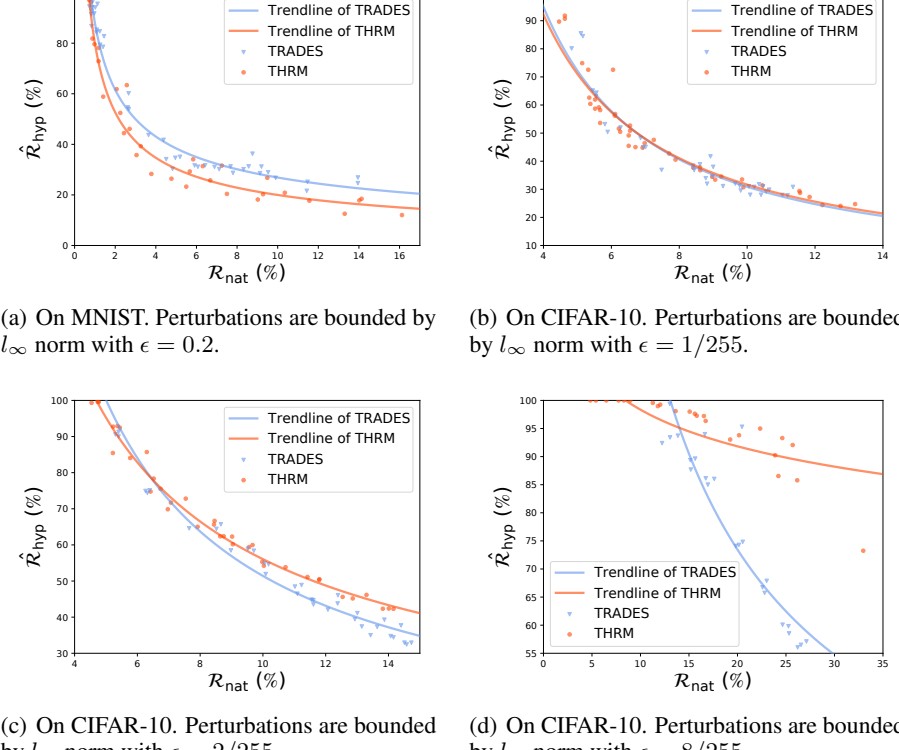

(a) On MNIST. Perturbations are bounded by $l_\infty$ norm with $\epsilon = 0.2$.

(b) On CIFAR-10. Perturbations are bounded by $l_\infty$ norm with $\epsilon = 1/255$.

(c) On CIFAR-10. Perturbations are bounded by $l_\infty$ norm with $\epsilon = 2/255$.

(d) On CIFAR-10. Perturbations are bounded by $l_\infty$ norm with $\epsilon = 8/255$.

Figure 6: Tradeoff between natural risk and hypocritical risk on real-world datasets.

Table 8: Full results of natural risk (%) and hypocritical risk (%) on MNIST. Attacks are bounded by $l_\infty$ norm with $\epsilon = 0.2$.

(a) For TRADES.

| $\lambda$ | Trial 1 | | Trial 2 | | Trial 3 | |
|---|---|---|---|---|---|---|
| | $\mathcal{R}_{nat}$ | $\hat{\mathcal{R}}_{hyp}$ | $\mathcal{R}_{nat}$ | $\hat{\mathcal{R}}_{hyp}$ | $\mathcal{R}_{nat}$ | $\hat{\mathcal{R}}_{hyp}$ |
| 0.1 | 0.7 | 94.5 | 0.9 | 94.2 | 0.9 | 96.7 |
| 1 | 1.0 | 94.1 | 0.8 | 91.5 | 0.8 | 92.8 |
| 5 | 0.8 | 86.7 | 1.0 | 91.8 | 1.1 | 95.6 |
| 10 | 1.2 | 84.7 | 1.1 | 85.3 | 1.1 | 84.3 |
| 20 | 1.5 | 82.8 | 1.4 | 78.6 | 1.3 | 79.4 |
| 40 | 2.7 | 60.2 | 2.7 | 53.9 | 2.6 | 54.6 |
| 60 | 4.4 | 41.8 | 4.5 | 34.1 | 3.6 | 43.7 |
| 80 | 5.2 | 35.1 | 5.0 | 34.7 | 4.8 | 30.3 |
| 120 | 6.1 | 31.1 | 6.9 | 31.1 | 5.9 | 31.6 |
| 160 | 7.8 | 28.7 | 8.6 | 31.3 | 6.4 | 31.7 |
| 200 | 9.2 | 31.2 | 7.7 | 31.3 | 7.1 | 30.2 |
| 240 | 9.9 | 26.8 | 8.8 | 36.3 | 9.1 | 28.6 |
| 300 | 8.3 | 30.2 | 14.0 | 26.9 | 9.5 | 29.0 |
| 500 | 11.4 | 21.6 | 14.0 | 24.7 | 11.5 | 25.2 |

(b) For THRM.

| $\lambda$ | Trial 1 | | Trial 2 | | Trial 3 | |
|---|---|---|---|---|---|---|
| | $\mathcal{R}_{nat}$ | $\hat{\mathcal{R}}_{hyp}$ | $\mathcal{R}_{nat}$ | $\hat{\mathcal{R}}_{hyp}$ | $\mathcal{R}_{nat}$ | $\hat{\mathcal{R}}_{hyp}$ |
| 1 | 0.5 | 100.0 | 0.6 | 100.0 | 0.7 | 100.0 |
| 10 | 0.7 | 100.0 | 0.8 | 100.0 | 0.7 | 100.0 |
| 100 | 0.7 | 100.0 | 0.7 | 98.6 | 0.7 | 100.0 |
| 200 | 0.8 | 96.3 | 0.7 | 97.2 | 0.7 | 98.5 |
| 400 | 1.0 | 79.6 | 1.2 | 72.9 | 0.9 | 81.8 |
| 600 | 2.1 | 61.8 | 1.4 | 58.9 | 1.2 | 78.2 |
| 800 | 2.7 | 46.1 | 2.6 | 63.4 | 2.4 | 44.4 |
| 1000 | 3.3 | 39.3 | 3.1 | 35.7 | 2.3 | 52.4 |
| 2000 | 5.8 | 34.1 | 4.8 | 26.4 | 3.8 | 28.3 |
| 3000 | 7.3 | 31.6 | 14.0 | 17.8 | 7.5 | 20.4 |
| 4000 | 6.7 | 25.7 | 5.5 | 23.3 | 9.0 | 18.2 |
| 5000 | 10.4 | 20.9 | 5.7 | 29.3 | 11.6 | 17.7 |
| 6000 | 14.1 | 18.5 | 16.1 | 12.0 | 9.5 | 26.8 |
| 7000 | 13.3 | 12.6 | 6.3 | 31.4 | 9.3 | 20.3 |

Table 9: Full results of natural risk (%) and hypocritical risk (%) on CIFAR-10. Attacks are bounded by $l_\infty$ norm with $\epsilon = 1/255$.

(a) For TRADES.

| $\lambda$ | Trial 1 | | Trial 2 | | Trial 3 | |
|---|---|---|---|---|---|---|
| | $\mathcal{R}_{nat}$ | $\hat{\mathcal{R}}_{hyp}$ | $\mathcal{R}_{nat}$ | $\hat{\mathcal{R}}_{hyp}$ | $\mathcal{R}_{nat}$ | $\hat{\mathcal{R}}_{hyp}$ |
| 0.1 | 5.2 | 84.5 | 5.1 | 85.5 | 4.8 | 80.1 |
| 1 | 5.6 | 63.1 | 5.5 | 65.1 | 5.6 | 64.4 |
| 5 | 6.3 | 52.0 | 5.9 | 50.4 | 5.8 | 53.2 |
| 10 | 6.9 | 48.3 | 6.9 | 45.0 | 7.0 | 45.1 |
| 30 | 9.0 | 38.0 | 8.9 | 41.7 | 7.5 | 36.9 |
| 50 | 8.6 | 38.9 | 8.4 | 36.8 | 8.4 | 38.4 |
| 70 | 9.0 | 34.5 | 8.8 | 33.9 | 8.8 | 32.0 |
| 90 | 9.7 | 32.0 | 10.4 | 28.0 | 9.3 | 31.1 |
| 120 | 9.8 | 31.3 | 9.8 | 29.6 | 10.7 | 29.9 |
| 150 | 10.2 | 30.9 | 10.1 | 28.0 | 10.1 | 31.0 |
| 180 | 10.6 | 29.2 | 10.4 | 31.8 | 10.5 | 31.0 |
| 210 | 11.2 | 27.9 | 11.4 | 30.8 | 11.0 | 28.0 |

(b) For THRM.

| $\lambda$ | Trial 1 | | Trial 2 | | Trial 3 | |
|---|---|---|---|---|---|---|
| | $\mathcal{R}_{nat}$ | $\hat{\mathcal{R}}_{hyp}$ | $\mathcal{R}_{nat}$ | $\hat{\mathcal{R}}_{hyp}$ | $\mathcal{R}_{nat}$ | $\hat{\mathcal{R}}_{hyp}$ |
| 1 | 4.6 | 91.8 | 4.6 | 90.7 | 4.5 | 89.7 |
| 10 | 5.1 | 74.9 | 6.1 | 72.6 | 5.3 | 72.6 |
| 50 | 5.3 | 62.6 | 5.5 | 62.0 | 5.4 | 60.4 |
| 100 | 5.6 | 59.1 | 5.7 | 58.2 | 5.5 | 58.7 |
| 200 | 6.6 | 52.7 | 6.5 | 49.2 | 6.2 | 51.6 |
| 300 | 6.9 | 44.8 | 6.5 | 45.5 | 7.0 | 46.4 |
| 400 | 7.3 | 47.6 | 7.7 | 42.7 | 7.9 | 40.6 |
| 600 | 8.4 | 37.7 | 8.6 | 36.8 | 8.6 | 37.9 |
| 800 | 9.1 | 33.4 | 9.2 | 34.5 | 9.0 | 34.4 |
| 1000 | 9.9 | 30.7 | 9.8 | 33.5 | 10.5 | 31.4 |
| 1500 | 11.6 | 28.7 | 11.5 | 29.4 | 11.8 | 27.3 |
| 2000 | 13.2 | 24.7 | 12.2 | 24.4 | 12.8 | 24.1 |

Table 10: Full results of natural risk (%) and hypocritical risk (%) on CIFAR-10. Attacks are bounded by $l_\infty$ norm with $\epsilon = 2/255$.

(a) For TRADES.

| $\lambda$ | Trial 1 | | Trial 2 | | Trial 3 | |
|---|---|---|---|---|---|---|
| | $\mathcal{R}_{nat}$ | $\hat{\mathcal{R}}_{hyp}$ | $\mathcal{R}_{nat}$ | $\hat{\mathcal{R}}_{hyp}$ | $\mathcal{R}_{nat}$ | $\hat{\mathcal{R}}_{hyp}$ |
| 0.1 | 5.4 | 91.3 | 5.3 | 90.6 | 5.4 | 90.0 |
| 1 | 6.3 | 74.4 | 6.3 | 74.9 | 6.4 | 75.1 |
| 5 | 8.5 | 64.4 | 7.7 | 64.6 | 8.7 | 65.7 |
| 10 | 9.7 | 58.4 | 9.0 | 58.5 | 9.5 | 58.9 |
| 30 | 10.2 | 54.6 | 10.1 | 51.9 | 10.1 | 54.9 |
| 50 | 11.1 | 46.5 | 11.2 | 48.9 | 11.0 | 48.5 |
| 70 | 11.6 | 44.7 | 11.6 | 43.5 | 11.6 | 44.8 |
| 90 | 12.1 | 42.1 | 12.4 | 46.1 | 12.4 | 43.8 |
| 120 | 13.2 | 37.4 | 13.0 | 41.1 | 12.9 | 39.4 |
| 150 | 13.6 | 37.3 | 13.9 | 39.3 | 13.4 | 35.0 |
| 180 | 14.1 | 34.8 | 14.2 | 34.4 | 14.4 | 37.7 |
| 210 | 14.7 | 32.9 | 14.6 | 32.4 | 14.5 | 32.9 |

(b) For THRM.

| $\lambda$ | Trial 1 | | Trial 2 | | Trial 3 | |
|---|---|---|---|---|---|---|
| | $\mathcal{R}_{nat}$ | $\hat{\mathcal{R}}_{hyp}$ | $\mathcal{R}_{nat}$ | $\hat{\mathcal{R}}_{hyp}$ | $\mathcal{R}_{nat}$ | $\hat{\mathcal{R}}_{hyp}$ |
| 1 | 4.8 | 99.6 | 4.5 | 99.3 | 4.7 | 99.6 |
| 10 | 5.4 | 92.9 | 5.4 | 92.5 | 5.2 | 92.7 |
| 50 | 6.3 | 85.7 | 5.8 | 84.1 | 5.2 | 85.4 |
| 100 | 6.7 | 75.5 | 6.4 | 74.8 | 6.5 | 78.4 |
| 150 | 7.1 | 71.7 | 7.0 | 69.9 | 7.5 | 72.8 |
| 200 | 8.4 | 65.6 | 7.9 | 65.0 | 8.5 | 66.7 |
| 250 | 8.8 | 62.4 | 8.6 | 62.4 | 9.0 | 62.3 |
| 300 | 9.6 | 59.4 | 9.0 | 60.2 | 9.7 | 60.0 |
| 400 | 10.0 | 54.2 | 10.0 | 55.3 | 10.7 | 53.8 |
| 600 | 11.8 | 50.5 | 11.4 | 51.1 | 11.8 | 50.3 |
| 800 | 12.5 | 45.6 | 12.9 | 45.2 | 13.3 | 46.2 |
| 1000 | 14.2 | 42.3 | 13.8 | 42.3 | 14.0 | 42.4 |

Table 11: Full results of natural risk (%) and hypocritical risk (%) on CIFAR-10. Attacks are bounded by $l_\infty$ norm with $\epsilon = 8/255$.

(a) For TRADES.

| $\lambda$ | Trial 1 | | Trial 2 | | Trial 3 | |
|---|---|---|---|---|---|---|
| | $\mathcal{R}_{nat}$ | $\hat{\mathcal{R}}_{hyp}$ | $\mathcal{R}_{nat}$ | $\hat{\mathcal{R}}_{hyp}$ | $\mathcal{R}_{nat}$ | $\hat{\mathcal{R}}_{hyp}$ |
| 0.1 | 16.6 | 94.0 | 20.5 | 95.3 | 13.0 | 99.4 |
| 1 | 13.9 | 93.7 | 13.1 | 93.4 | 12.2 | 92.4 |
| 5 | 15.1 | 89.4 | 15.2 | 87.7 | 15.6 | 89.6 |
| 10 | 17.0 | 85.0 | 17.6 | 86.0 | 16.7 | 86.1 |
| 30 | 19.8 | 74.1 | 20.2 | 74.3 | 20.5 | 74.8 |
| 50 | 23.0 | 67.9 | 22.8 | 65.7 | 22.6 | 66.7 |
| 70 | 25.3 | 59.8 | 24.7 | 60.1 | 25.3 | 58.5 |
| 90 | 26.6 | 56.5 | 27.1 | 57.1 | 26.2 | 56.0 |

(b) For THRM.

| $\lambda$ | Trial 1 | | Trial 2 | | Trial 3 | |
|---|---|---|---|---|---|---|
| | $\mathcal{R}_{nat}$ | $\hat{\mathcal{R}}_{hyp}$ | $\mathcal{R}_{nat}$ | $\hat{\mathcal{R}}_{hyp}$ | $\mathcal{R}_{nat}$ | $\hat{\mathcal{R}}_{hyp}$ |
| 1 | 4.8 | 100.0 | 9.0 | 100.0 | 5.4 | 100.0 |
| 10 | 8.3 | 100.0 | 6.5 | 100.0 | 7.7 | 100.0 |
| 50 | 12.0 | 99.3 | 11.3 | 99.6 | 11.8 | 99.0 |
| 100 | 15.1 | 98.0 | 13.6 | 98.1 | 15.6 | 97.6 |
| 150 | 16.7 | 96.4 | 15.8 | 97.3 | 16.5 | 97.2 |
| 200 | 19.3 | 93.0 | 20.2 | 93.8 | 22.3 | 95.0 |
| 250 | 24.2 | 86.5 | 23.9 | 90.2 | 24.6 | 93.3 |
| 300 | 25.7 | 92.1 | 33.0 | 73.2 | 26.2 | 85.8 |

Table 12: Comparison of natural risk (%±std over 3 random trials) and hypocritical risk (%±std over 3 random trials) between methods on real-world datasets. Attacks are bounded by $l_\infty$ norm.

(a) On MNIST ($\epsilon = 0.2$).

| Method | $\mathcal{R}_{nat}$ | $\hat{\mathcal{R}}_{hyp}$ |
|---|---|---|
| Standard | $0.6 \pm 0.1$ | $100.0 \pm 0.0$ |
| Madry | $0.9 \pm 0.0$ | $93.8 \pm 1.5$ |
| TRADES ($\lambda = 240$) | $9.3 \pm 0.6$ | $30.6 \pm 5.1$ |
| THRM ($\lambda = 5000$) | $9.2 \pm 3.1$ | $22.6 \pm 6.0$ |

(b) On CIFAR-10 ($\epsilon = 1/255$).

| Method | $\mathcal{R}_{nat}$ | $\hat{\mathcal{R}}_{hyp}$ |
|---|---|---|
| Standard | $4.9 \pm 0.2$ | $99.3 \pm 0.3$ |
| Madry | $5.3 \pm 0.1$ | $62.6 \pm 0.1$ |
| TRADES ($\lambda = 150$) | $10.1 \pm 0.1$ | $30.0 \pm 1.7$ |
| THRM ($\lambda = 1000$) | $10.1 \pm 0.3$ | $31.9 \pm 1.4$ |

(c) On CIFAR-10 ($\epsilon = 2/255$).

| Method | $\mathcal{R}_{nat}$ | $\hat{\mathcal{R}}_{hyp}$ |
|---|---|---|
| Standard | $4.8 \pm 0.2$ | $100 \pm 0.0$ |
| Madry | $6.3 \pm 0.2$ | $74.3 \pm 1.0$ |
| TRADES ($\lambda = 150$) | $13.6 \pm 0.2$ | $37.2 \pm 2.2$ |
| THRM ($\lambda = 1000$) | $14.0 \pm 0.2$ | $42.3 \pm 0.0$ |

(d) On CIFAR-10 ($\epsilon = 8/255$).

| Method | $\mathcal{R}_{nat}$ | $\hat{\mathcal{R}}_{hyp}$ |
|---|---|---|
| Standard | $4.7 \pm 0.3$ | $100.0 \pm 0.0$ |
| Madry | $14.2 \pm 0.1$ | $93.1 \pm 1.0$ |
| TRADES ($\lambda = 50$) | $22.8 \pm 0.2$ | $66.8 \pm 1.1$ |
| THRM ($\lambda = 250$) | $24.2 \pm 0.4$ | $90.0 \pm 3.4$ |

Table 13: Evaluated results of natural risk (%) and adversarial risk (%) on MNIST. Attacks are bounded by $l_\infty$ norm with $\epsilon = 0.2$.

(a) For TRADES.

| $\lambda$ | Trial 1 | | Trial 2 | | Trial 3 | |
|---|---|---|---|---|---|---|
| | $\mathcal{R}_{nat}$ | $\hat{\mathcal{R}}_{adv}$ | $\mathcal{R}_{nat}$ | $\hat{\mathcal{R}}_{adv}$ | $\mathcal{R}_{nat}$ | $\hat{\mathcal{R}}_{adv}$ |
| 0.1 | 0.7 | 5.9 | 0.9 | 6.9 | 0.9 | 6.2 |
| 1 | 1.0 | 5.7 | 0.8 | 5.6 | 0.8 | 5.2 |
| 5 | 0.8 | 4.4 | 1.0 | 5.4 | 1.1 | 5.0 |
| 10 | 1.2 | 4.3 | 1.1 | 4.0 | 1.1 | 4.4 |
| 20 | 1.5 | 3.6 | 1.4 | 4.7 | 1.3 | 4.3 |
| 40 | 2.7 | 3.2 | 2.7 | 3.2 | 2.6 | 3.2 |
| 60 | 4.4 | 3.5 | 4.5 | 3.0 | 3.6 | 3.5 |
| 80 | 5.2 | 2.9 | 5.0 | 3.3 | 4.8 | 3.4 |
| 120 | 6.1 | 3.5 | 6.9 | 4.4 | 5.9 | 3.8 |
| 160 | 7.8 | 4.7 | 8.6 | 4.9 | 6.4 | 4.0 |
| 200 | 9.2 | 5.1 | 7.7 | 4.9 | 7.1 | 4.1 |
| 240 | 9.9 | 4.7 | 8.8 | 5.2 | 9.1 | 4.7 |
| 300 | 8.3 | 5.1 | 14.0 | 7.1 | 9.5 | 4.7 |
| 500 | 11.4 | 4.8 | 14.0 | 5.8 | 11.5 | 5.6 |

(b) For THRM.

| $\lambda$ | Trial 1 | | Trial 2 | | Trial 3 | |
|---|---|---|---|---|---|---|
| | $\mathcal{R}_{nat}$ | $\hat{\mathcal{R}}_{adv}$ | $\mathcal{R}_{nat}$ | $\hat{\mathcal{R}}_{adv}$ | $\mathcal{R}_{nat}$ | $\hat{\mathcal{R}}_{adv}$ |
| 1 | 0.5 | 100.0 | 0.6 | 99.8 | 0.7 | 100.0 |
| 10 | 0.7 | 100.0 | 0.8 | 99.8 | 0.7 | 99.9 |
| 100 | 0.7 | 92.2 | 0.7 | 94.8 | 0.7 | 91.7 |
| 200 | 0.8 | 86.0 | 0.7 | 82.1 | 0.7 | 85.7 |
| 400 | 1.0 | 67.1 | 1.2 | 59.1 | 0.9 | 82.7 |
| 600 | 2.1 | 39.9 | 1.4 | 37.2 | 1.2 | 73.6 |
| 800 | 2.7 | 35.3 | 2.6 | 28.8 | 2.4 | 61.0 |
| 1000 | 3.3 | 23.6 | 3.1 | 24.7 | 2.3 | 51.5 |
| 2000 | 5.8 | 21.6 | 4.8 | 17.3 | 3.8 | 23.8 |
| 3000 | 7.3 | 31.4 | 14.0 | 30.0 | 7.5 | 29.4 |
| 4000 | 6.7 | 39.2 | 5.5 | 29.2 | 9.0 | 23.9 |
| 5000 | 10.4 | 29.5 | 5.7 | 25.6 | 11.6 | 20.8 |
| 6000 | 14.1 | 53.4 | 16.1 | 21.1 | 9.5 | 23.9 |
| 7000 | 13.3 | 31.9 | 6.3 | 22.8 | 9.3 | 29.0 |

Table 14: Evaluated results of natural risk (%) and adversarial risk (%) on CIFAR-10. Attacks are bounded by $l_\infty$ norm with $\epsilon = 2/255$.

(a) For TRADES.

| $\lambda$ | Trial 1 | | Trial 2 | | Trial 3 | |
|---|---|---|---|---|---|---|
| | $\mathcal{R}_{nat}$ | $\hat{\mathcal{R}}_{adv}$ | $\mathcal{R}_{nat}$ | $\hat{\mathcal{R}}_{adv}$ | $\mathcal{R}_{nat}$ | $\hat{\mathcal{R}}_{adv}$ |
| 0.1 | 5.4 | 21.7 | 5.3 | 24.3 | 5.4 | 24.1 |
| 1 | 6.3 | 13.3 | 6.3 | 13.6 | 6.4 | 13.3 |
| 5 | 8.5 | 12.1 | 7.7 | 11.0 | 8.7 | 12.5 |
| 10 | 9.7 | 11.7 | 9.0 | 10.7 | 9.5 | 11.1 |
| 30 | 10.2 | 9.8 | 10.1 | 9.5 | 10.1 | 9.7 |
| 50 | 11.1 | 9.0 | 11.2 | 9.2 | 11.0 | 8.7 |
| 70 | 11.6 | 9.0 | 11.6 | 8.9 | 11.6 | 8.6 |
| 90 | 12.1 | 8.9 | 12.4 | 8.3 | 12.4 | 9.2 |
| 120 | 13.2 | 8.3 | 13.0 | 7.8 | 12.9 | 8.0 |
| 150 | 13.6 | 8.0 | 13.9 | 8.4 | 13.4 | 7.7 |
| 180 | 14.1 | 8.4 | 14.2 | 7.4 | 14.4 | 7.8 |
| 210 | 14.7 | 7.9 | 14.6 | 7.5 | 14.5 | 7.8 |

(b) For THRM.

| $\lambda$ | Trial 1 | | Trial 2 | | Trial 3 | |
|---|---|---|---|---|---|---|
| | $\mathcal{R}_{nat}$ | $\hat{\mathcal{R}}_{adv}$ | $\mathcal{R}_{nat}$ | $\hat{\mathcal{R}}_{adv}$ | $\mathcal{R}_{nat}$ | $\hat{\mathcal{R}}_{adv}$ |
| 1 | 4.8 | 55.7 | 4.5 | 57.5 | 4.7 | 53.1 |
| 10 | 5.4 | 26.8 | 5.4 | 27.6 | 5.2 | 28.2 |
| 50 | 6.3 | 18.9 | 5.8 | 18.5 | 5.2 | 19.2 |
| 100 | 6.7 | 15.1 | 6.4 | 14.8 | 6.5 | 15.3 |
| 150 | 7.1 | 13.9 | 7.0 | 14.0 | 7.5 | 14.1 |
| 200 | 8.4 | 12.4 | 7.9 | 13.4 | 8.5 | 13.3 |
| 250 | 8.8 | 12.2 | 8.6 | 12.1 | 9.0 | 12.8 |
| 300 | 9.6 | 12.1 | 9.0 | 11.9 | 9.7 | 11.5 |
| 400 | 10.0 | 11.3 | 10.0 | 11.6 | 10.7 | 11.0 |
| 600 | 11.8 | 11.2 | 11.4 | 11.2 | 11.8 | 10.4 |
| 800 | 12.5 | 10.5 | 12.9 | 10.7 | 13.3 | 10.3 |
| 1000 | 14.2 | 10.7 | 13.8 | 10.5 | 14.0 | 10.1 |

## A.4 DETAILS IN SECTION 5

Hypocritical attacks here are executed by 50 steps PGD (step size is $\epsilon/10$) on source models. Note that the optimization method we used here is not to pursue state-of-the-art transferability, but to examine the transferability of hypocritical examples. There are many methods designed to improve the transferability of adversarial examples may be extended to hypocritical examples (Liu et al., 2017; Dong et al., 2018; Wu et al., 2020a). Figure 7 shows the transferability heatmap of hypocritical attack over 9 models trained on MNIST. Figure 8 shows the transferability heatmap of hypocritical attack over 7 models trained on CIFAR-10. The value in the $i$-th row and $j$-th clumn of each heatmap matrix is the proportion of the hypocritical examples successfully transferred to target model $j$ out of all hypocritical examples generated by source model $i$ (including both successful and failed attacks on the source model).

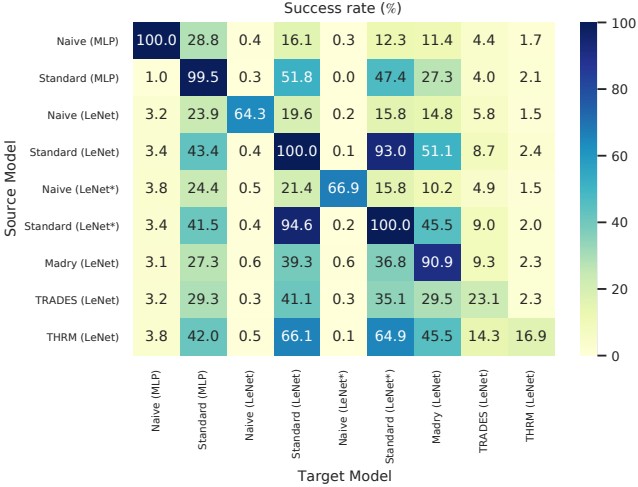

Figure 7: Transferability of hypocritical examples on MNIST. Attacks are bounded by $l_\infty$ norm with $\epsilon = 0.2$. "(LeNet*)" means that it is the same architecture with "(LeNet)" but different random initialization.

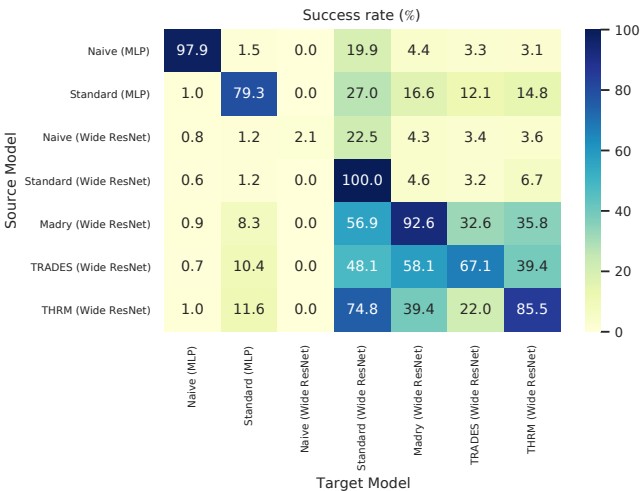

Figure 8: Transferability of hypocritical examples on CIFAR-10. Attacks are bounded by $l_\infty$ norm with $\epsilon = 8/255$.

# B  PROOFS OF MAIN RESULTS

In this section, we provide the proofs of our main results.

## B.1  A PROOF OF PROPOSITION 1

**Proposition 1.** *Denote the adversarial risk on correctly classified examples by*

$$\hat{\mathcal{R}}_{\mathsf{adv}}(f) = \mathop{\mathbb{E}}_{(\boldsymbol{x},y)\sim\mathcal{S}_f^+} \left[ \max_{\boldsymbol{x}'\in\mathcal{B}_\epsilon(\boldsymbol{x})} \mathbb{1}(f(\boldsymbol{x}') \neq y) \right],$$

*then we have*

$$\mathcal{R}_{\mathsf{adv}}(f) = \mathcal{R}_{\mathsf{nat}}(f) + (1 - \mathcal{R}_{\mathsf{nat}}(f))\hat{\mathcal{R}}_{\mathsf{adv}}(f).$$

*Proof.*

$$\begin{aligned}
\mathcal{R}_{\mathsf{adv}}(f) &= \mathop{\mathbb{E}}_{(\boldsymbol{x},y)\sim\mathcal{D}} \left[ \max_{\boldsymbol{x}'\in\mathcal{B}_\epsilon(\boldsymbol{x})} \mathbb{1}(f(\boldsymbol{x}') \neq y) \right] \\
&= \mathop{\mathbb{E}}_{(\boldsymbol{x},y)\sim\mathcal{D}} \left[ \mathbb{1}(f(\boldsymbol{x}) = y) \cdot \max_{\boldsymbol{x}'\in\mathcal{B}_\epsilon(\boldsymbol{x})} \mathbb{1}(f(\boldsymbol{x}') \neq y) \right] \\
&\quad + \mathop{\mathbb{E}}_{(\boldsymbol{x},y)\sim\mathcal{D}} \left[ \mathbb{1}(f(\boldsymbol{x}) \neq y) \cdot \max_{\boldsymbol{x}'\in\mathcal{B}_\epsilon(\boldsymbol{x})} \mathbb{1}(f(\boldsymbol{x}') \neq y) \right] \\
&= \mathcal{R}_{\mathsf{nat}}(f) \mathop{\mathbb{E}}_{(\boldsymbol{x},y)\sim\mathcal{S}_f^-} \left[ \max_{\boldsymbol{x}'\in\mathcal{B}_\epsilon(\boldsymbol{x})} \mathbb{1}(f(\boldsymbol{x}') \neq y) \right] \\
&\quad + (1 - \mathcal{R}_{\mathsf{nat}}(f)) \mathop{\mathbb{E}}_{(\boldsymbol{x},y)\sim\mathcal{S}_f^+} \left[ \max_{\boldsymbol{x}'\in\mathcal{B}_\epsilon(\boldsymbol{x})} \mathbb{1}(f(\boldsymbol{x}') \neq y) \right] \\
&= \mathcal{R}_{\mathsf{nat}}(f) \mathop{\mathbb{E}}_{(\boldsymbol{x},y)\sim\mathcal{S}_f^-} \left[ \mathbb{1}(f(\boldsymbol{x}) \neq y) \right] + (1 - \mathcal{R}_{\mathsf{nat}}(f)) \mathop{\mathbb{E}}_{(\boldsymbol{x},y)\sim\mathcal{S}_f^+} \left[ \max_{\boldsymbol{x}'\in\mathcal{B}_\epsilon(\boldsymbol{x})} \mathbb{1}(f(\boldsymbol{x}') \neq y) \right] \\
&= \mathcal{R}_{\mathsf{nat}}(f) + (1 - \mathcal{R}_{\mathsf{nat}}(f)) \mathop{\mathbb{E}}_{(\boldsymbol{x},y)\sim\mathcal{S}_f^+} \left[ \max_{\boldsymbol{x}'\in\mathcal{B}_\epsilon(\boldsymbol{x})} \mathbb{1}(f(\boldsymbol{x}') \neq y) \right] \\
&= \mathcal{R}_{\mathsf{nat}}(f) + (1 - \mathcal{R}_{\mathsf{nat}}(f))\hat{\mathcal{R}}_{\mathsf{adv}}(f).
\end{aligned}$$

$\square$

## B.2  A PROOF OF THEOREM 1

**Theorem 1.** *For any data distribution $\mathcal{D}$ and its corresponding conditional distribution on misclassified examples $\mathcal{S}_f^-$ w.r.t. a classifier $f$, we have*

$$\underbrace{\mathop{\mathbb{E}}_{(\boldsymbol{x},y)\sim\mathcal{S}_f^-} \mathbb{1}(f(\boldsymbol{x}_{\mathsf{hyp}}) = y)}_{\hat{\mathcal{R}}_{\mathsf{hyp}}(f)} \leq \underbrace{\mathop{\mathbb{E}}_{(\boldsymbol{x},y)\sim\mathcal{S}_f^-} \mathbb{1}(f(\boldsymbol{x}_{\mathsf{hyp}}) \neq f(\boldsymbol{x}))}_{\overline{\mathcal{R}}_{\mathsf{hyp}}(f)} \leq \underbrace{\mathop{\mathbb{E}}_{(\boldsymbol{x},y)\sim\mathcal{S}_f^-} \mathbb{1}(f(\boldsymbol{x}_{\mathsf{rev}}) \neq f(\boldsymbol{x}))}_{\overline{\overline{\mathcal{R}}}_{\mathsf{hyp}}(f)},$$

*where $\boldsymbol{x}_{\mathsf{hyp}} = \arg\max_{\boldsymbol{x}'\in\mathcal{B}_\epsilon(\boldsymbol{x})} \mathbb{1}(f(\boldsymbol{x}') = y)$ and $\boldsymbol{x}_{\mathsf{rev}} = \arg\max_{\boldsymbol{x}'\in\mathcal{B}_\epsilon(\boldsymbol{x})} \mathbb{1}(f(\boldsymbol{x}') \neq f(\boldsymbol{x}))$.*

*Proof.* To prove the first inequality, we have

$$\begin{aligned}
\hat{\mathcal{R}}_{\mathsf{hyp}}(f) &= \mathop{\mathbb{E}}_{(\boldsymbol{x},y)\sim\mathcal{S}_f^-} \left[ \max_{\boldsymbol{x}'\in\mathcal{B}_\epsilon(\boldsymbol{x})} \mathbb{1}(f(\boldsymbol{x}') = y) \right] \\
&= \mathop{\mathbb{E}}_{(\boldsymbol{x},y)\sim\mathcal{S}_f^-} \mathbb{1}(f(\boldsymbol{x}_{\mathsf{hyp}}) = y) \\
&\leq \mathop{\mathbb{E}}_{(\boldsymbol{x},y)\sim\mathcal{S}_f^-} \mathbb{1}(f(\boldsymbol{x}_{\mathsf{hyp}}) \neq f(\boldsymbol{x})),
\end{aligned}$$

where the above inequality involves two conditions:

$$\mathbb{1}(f(\boldsymbol{x}_{\mathsf{hyp}}) = y) = \left\{ \begin{array}{ll} 1 = \mathbb{1}(f(\boldsymbol{x}_{\mathsf{hyp}}) \neq f(\boldsymbol{x})), & \text{if } f(\boldsymbol{x}_{\mathsf{hyp}}) = y, \\ 0 \leq \mathbb{1}(f(\boldsymbol{x}_{\mathsf{hyp}}) \neq f(\boldsymbol{x})), & \text{if } f(\boldsymbol{x}_{\mathsf{hyp}}) \neq y. \end{array} \right.$$

To prove the second inequality, we have

$$\overline{\mathcal{R}}_{\mathsf{hyp}}(f) = \mathop{\mathbb{E}}_{(\boldsymbol{x},y)\sim\mathcal{S}_f^-} \mathbb{1}(f(\boldsymbol{x}_{\mathsf{hyp}}) \neq f(\boldsymbol{x}))$$
$$\leq \mathop{\mathbb{E}}_{(\boldsymbol{x},y)\sim\mathcal{S}_f^-} \mathbb{1}(f(\boldsymbol{x}_{\mathsf{rev}}) \neq f(\boldsymbol{x})).$$

Since $(\boldsymbol{x},y) \sim \mathcal{S}_f^-$, we have $f(\boldsymbol{x}) \neq y$. If there exists a $\boldsymbol{x}_{\mathsf{hyp}}$ such that $f(\boldsymbol{x}_{\mathsf{hyp}}) = y$, then $f(\boldsymbol{x}_{\mathsf{hyp}}) \neq f(\boldsymbol{x})$. Now let $\boldsymbol{x}_{\mathsf{rev}} = \boldsymbol{x}_{\mathsf{hyp}}$, then $f(\boldsymbol{x}_{\mathsf{rev}}) \neq f(\boldsymbol{x})$ is true. Otherwise, if we couldn't find a $\boldsymbol{x}_{\mathsf{hyp}}$ such that $f(\boldsymbol{x}_{\mathsf{hyp}}) = y$, there still exists a posibility to find a $\boldsymbol{x}_{\mathsf{rev}}$ such that $f(\boldsymbol{x}_{\mathsf{rev}}) \neq y$ but $f(\boldsymbol{x}_{\mathsf{rev}}) \neq f(\boldsymbol{x})$ is true. Therefore, the above inequalities holds. $\square$

### B.3 A Proof of Proposition 2

**Proposition 2.** $\mathcal{R}_{\mathsf{rev}}(f) = (1 - \mathcal{R}_{\mathsf{nat}}(f))\hat{\mathcal{R}}_{\mathsf{adv}}(f) + \mathcal{R}_{\mathsf{nat}}(f)\overline{\overline{\mathcal{R}}}_{\mathsf{hyp}}(f) = \mathop{\mathbb{E}}_{(\boldsymbol{x},y)\sim\mathcal{D}} \mathbb{1}(f(\boldsymbol{x}_{\mathsf{rev}}) \neq f(\boldsymbol{x})).$

*Proof.*

$$\mathcal{R}_{\mathsf{rev}}(f) = (1 - \mathcal{R}_{\mathsf{nat}}(f))\hat{\mathcal{R}}_{\mathsf{adv}}(f) + \mathcal{R}_{\mathsf{nat}}(f)\overline{\overline{\mathcal{R}}}_{\mathsf{hyp}}(f)$$

$$= (1 - \mathcal{R}_{\mathsf{nat}}(f)) \mathop{\mathbb{E}}_{(\boldsymbol{x},y)\sim\mathcal{S}_f^+} [\mathbb{1}(f(\boldsymbol{x}_{\mathsf{adv}}) \neq y)] + \mathcal{R}_{\mathsf{nat}}(f) \mathop{\mathbb{E}}_{(\boldsymbol{x},y)\sim\mathcal{S}_f^-} [\mathbb{1}(f(\boldsymbol{x}_{\mathsf{rev}}) \neq f(\boldsymbol{x}))]$$

$$= \mathop{\mathbb{E}}_{(\boldsymbol{x},y)\sim\mathcal{D}} [\mathbb{1}(f(\boldsymbol{x}) = y) \cdot \mathbb{1}(f(\boldsymbol{x}_{\mathsf{adv}}) \neq y)] + \mathop{\mathbb{E}}_{(\boldsymbol{x},y)\sim\mathcal{D}} [\mathbb{1}(f(\boldsymbol{x}) \neq y) \cdot \mathbb{1}(f(\boldsymbol{x}_{\mathsf{rev}}) \neq f(\boldsymbol{x}))]$$

$$= \mathop{\mathbb{E}}_{(\boldsymbol{x},y)\sim\mathcal{D}} \left[ \mathbb{1}(f(\boldsymbol{x}) = y) \cdot \max_{\boldsymbol{x}'\in\mathcal{B}_\epsilon(\boldsymbol{x})} \mathbb{1}(f(\boldsymbol{x}') \neq y) \right]$$

$$+ \mathop{\mathbb{E}}_{(\boldsymbol{x},y)\sim\mathcal{D}} \left[ \mathbb{1}(f(\boldsymbol{x}) \neq y) \cdot \max_{\boldsymbol{x}'\in\mathcal{B}_\epsilon(\boldsymbol{x})} \mathbb{1}(f(\boldsymbol{x}') \neq f(\boldsymbol{x})) \right]$$

$$= \mathop{\mathbb{E}}_{(\boldsymbol{x},y)\sim\mathcal{D}} \left[ \mathbb{1}(f(\boldsymbol{x}) = y) \cdot \max_{\boldsymbol{x}'\in\mathcal{B}_\epsilon(\boldsymbol{x})} \mathbb{1}(f(\boldsymbol{x}') \neq f(\boldsymbol{x})) \right]$$

$$+ \mathop{\mathbb{E}}_{(\boldsymbol{x},y)\sim\mathcal{D}} \left[ \mathbb{1}(f(\boldsymbol{x}) \neq y) \cdot \max_{\boldsymbol{x}'\in\mathcal{B}_\epsilon(\boldsymbol{x})} \mathbb{1}(f(\boldsymbol{x}') \neq f(\boldsymbol{x})) \right]$$

$$= \mathop{\mathbb{E}}_{(\boldsymbol{x},y)\sim\mathcal{D}} \left[ \max_{\boldsymbol{x}'\in\mathcal{B}_\epsilon(\boldsymbol{x})} \mathbb{1}(f(\boldsymbol{x}') \neq f(\boldsymbol{x})) \right]$$

$$= \mathop{\mathbb{E}}_{(\boldsymbol{x},y)\sim\mathcal{D}} [\mathbb{1}(f(\boldsymbol{x}_{\mathsf{rev}}) \neq f(\boldsymbol{x}))].$$

$\square$

## C Additional Discussions

We showed that correctly classified examples (hypocritical examples) could be easily found in the vicinity of misclassified clean examples. As a result, a hypocritically perturbed set could be constructed with these hypocritical examples. The victim model's standard accuracy evaluated on the hypocritically perturbed set becomes higher than that on the clean set. It is natural then to wonder: *How about adversarially robust accuracy (i.e., accuracy under adversarial perturbations) of the victim model on hypocritical examples?* It's easy to see that, if the adversary is bounded by the same $\epsilon$ ball as the false friend, the model's adversarial accuracy evaluated on hypocritically perturbed set is zero, since a misclassified example exists in the $\epsilon$ ball of a hypocritical example (by definition). However, if the adversary's power is restricted by another $\delta$ ball such that $\delta < \epsilon$, then a robust hypocritical example may exist in the vicinity of a clean example so that a $\delta$-bounded adversary can not change the model's prediction on the robust hypocritical example. In such a case, the model's adversarial accuracy evaluated on the robustly hypocritically perturbed set could be higher than that on the clean set. New attack and defense methods are required to further explore this phenomenon.

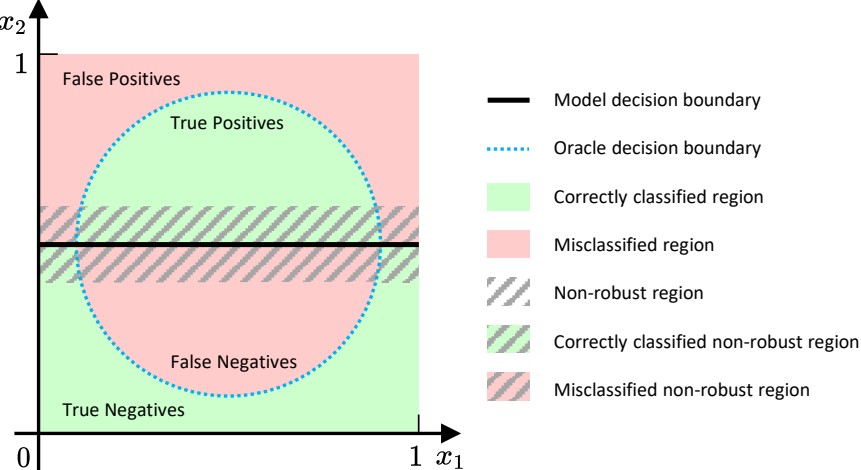

Figure 9: A visualization of the toy example to illustrate the phenomenon of the tradeoff between adversarial and hypocritical risks. Oracle decision boundary is the circle given by Equation 9. Model decision boundary is the line given by Equation 10 with the threshold $b = 0.5$. Red shadow and green shadow region denote that the points in there are misclassified and correctly classified by the model, respectively. The gray lined region denotes that the points in there can be perturbed with little perturbations to reverse the prediction of the model.

## D  TRADEOFF BETWEEN ADVERSARIAL AND HYPOCRITICAL RISKS

Despite the experiments in Section 4.1 and Appendix A.3 showed that, when dealing with finite sample size and finite-time gradient-descent trained classifiers, there may be mutual benefits between adversarial robustness and hypocritical robustness in real-world datasets, we note that in general, this synergism does not necessarily exist. We illustrate the phenomenon by providing another toy example here, which is inspired by the precision-recall tradeoff Buckland & Gey (1994); Alvarez (2002).

Consider the case $(\boldsymbol{x}, y) \in \mathbb{R}^2 \times \{-1, +1\}$ from a distribution $\mathcal{D}$, where the marginal distribution over the instance space is a uniform distribution over $[0, 1]^2$. We assume that the decision boundary of the oracle (ground truth) is a circle:

$$\mathcal{O}(\boldsymbol{x}) = \text{sign}(r - \|\boldsymbol{x} - \boldsymbol{c}\|_2), \tag{9}$$

where the centre $\boldsymbol{c} = (0.5, 0.5)^\top$ and the radius $r = 0.4$. The points inside the circle are labeled as belonging to the positive class, otherwise they are labeled as belonging to the negative class. We consider the linear classifier $f$ with fixed $\boldsymbol{w} = (0, 1)^\top$ and a tunable threshold $b$:

$$f(\boldsymbol{x}) = \text{sign}(\boldsymbol{w}^\top \boldsymbol{x} - b) = \text{sign}(x_2 - b). \tag{10}$$

See Figure 9 for a visualization of the oracle and the linear classifier over the instance space. In this problem, we can show the tradeoffs by tuning the threshold $b$ of the linear classifier over $[0, 1]$. The precision is the number of true positives (i.e. the number of examples correctly classified as positive class) divided by the the sum of true positives and false positives (i.e. the number of examples misclassified as positive class). The recall is the number of true positives divided by the sum of true positives and false negatives (i.e. the number of examples misclassified as negative class). We compare the adversarial risk on correctly classified examples $\hat{\mathcal{R}}_{\text{adv}}(f)$ defined in Proposition 1 and the hypocritical risk on misclassified examples $\hat{\mathcal{R}}_{\text{hyp}}(f)$ defined in Definition 3. The computing

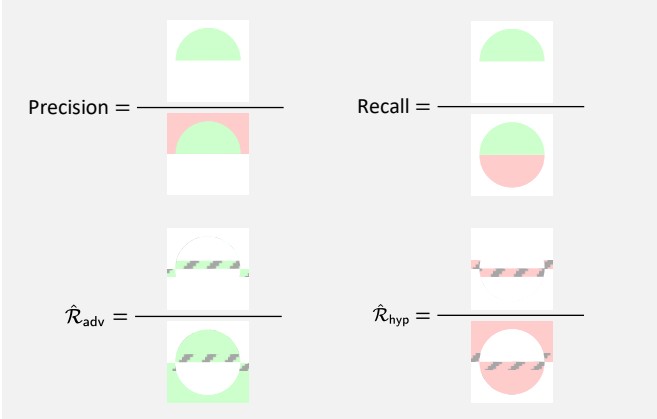

Figure 10: Visualization of the computing formulas of precision, recall, adversarial risk, and hypo-critical risk in the toy example. These values can be viewed as the proportion of the areas of different regions.

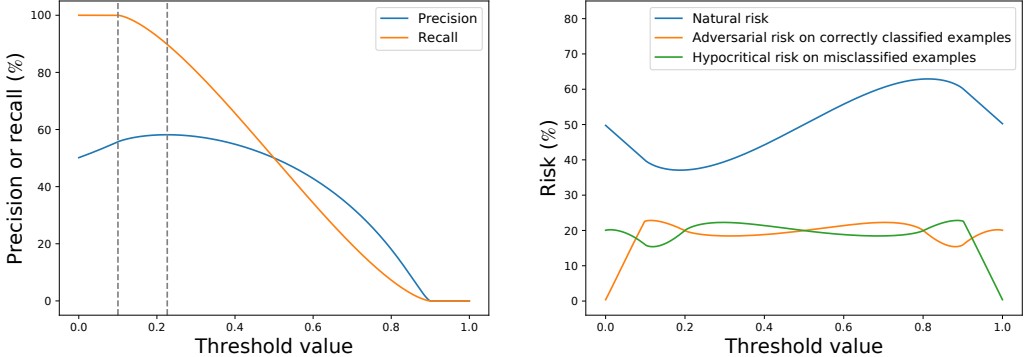

Figure 11: The tradeoff between precision and recall in the toy example.

Figure 12: The tradeoff between adversarial and hypocritical risks in the toy example.

formulas of these values are visualized in Figure 10. Here we choose the bounded $l_2$ ball $\mathcal{B}_\epsilon(\boldsymbol{x}) = \{\boldsymbol{x}' \in \mathbb{R}^2 : \|\boldsymbol{x}' - \boldsymbol{x}\|_2 \le \epsilon\}$ with $\epsilon = 0.1$ as the threat model.

Figure 11 plots the curve of precision and recall versus threshold $b$. We can see that there is a obvious precision-recall tradeoff between the two gray dotted lines. Similarly, Figure 12 plots the curve of $\hat{\mathcal{R}}_{\mathrm{adv}}(f)$ and $\hat{\mathcal{R}}_{\mathrm{hyp}}(f)$ versus threshold $b$. We can see that the tradeoff exists almost everywhere: as the adversarial risk increases, the hypocritical risk decreases, and vise versa.

