# OpenReview forum: "With False Friends Like These, Who Can Have Self-Knowledge?"
_ICLR.cc/2021/Conference — Reject_

### Official Review · AnonReviewer3 · 2020-10-28
**The main idea of the paper is decent, but there are many holes in the paper**

**Rating:** 4
**Confidence:** 5

**Review:**

This paper considers attacks that flip the label for those data points that are incorrectly classified. It proposes a risk metric to capture this and uses prior attacks for this type of attack.

Positive points:
The main idea in the paper is, at first look, interesting but then on more thinking I have doubts (see below):

Negative points:
- It is not very surprising that mis-classified data points can be perturbed to be properly classified using exactly the same kind of attack as prior work (not sure why authors call the attack conceptually different).
- I think the work is missing an important comparison to R_bdy (and R_rob) in Zhang et al. Is that in Propositon 2? Why call it R_rev?
- Also, why is equation (4) a natural risk to minimize? In Zhang et al., the term to minimize is nicely motivated by bounding R_rob - R^*_nat, which naturally leads to the tradeoff. I do not see how the tradeoff arises here.
- Also, the fact that TRADES would help here is obvious, isn't it? As TRADES handles two sided errors, it naturally also takes care of the one side of the error the authors care about. And that naturally raises the question that will only caring about one side of the error (in Eq 4) increase the other side (R_adv)? This R_adv is not shown in experiments. Also, shouldn't the experiment be done with larger and more variety of datasets - particularly, when for more complex datasets TRADES is doing reasonably well - so might be better off using TRADES for complex and real image datasets.
- A lot of the technically machinery is not very novel with same attack as before.

Minor points:
As a writing style, false friends is a good analogy but please do not use it throughout the paper in place of technical terms such as this line "Better transferability is beneficial for black-box attacks but is not always desired by false friends".

----------------Post rebuttal-----------
The issues I raised still persist in my mind. Also, other reviewers have similar issues and also more issues other than what I point out, which seem valid issues to me. I will keep my score as is.

---

> ### Author Response · Authors · 2020-11-18
> **Response to AnonReviewer3 (2/2)**
>
> **Question 6:** This R_adv is not shown in experiments.
>
> Thanks for your suggestion. Since we mainly cared about the tradeoff between natural risk and hypocritical risk in experiments, we did not evaluate the adversarial risk $\hat{R}_{\mathsf{adv}}$ of the trained models.
>
> For completeness, we further evaluate the $\hat{R}_{\mathsf{adv}}$. Results on MNIST ($\epsilon=0.2$) and CIFAR-10 ($\epsilon=2/255$) are summarized in Table 13 and Table 14 in the revision. We will add the results on CIFAR-10 ($\epsilon=1/255$ and $\epsilon=8/255$) in the final version. Below we summarize the average results on CIFAR-10 ($\epsilon=2/255$)  for your convenience.
>
> For TRADES:
>
> | $\lambda$                | 0.1   | 1     | 5     | 10    | 30    | 50    | 70    | 90    | 120   | 150   | 180   | 210   |
> | ------------------------ | ----- | ----- | ----- | ----- | ----- | ----- | ----- | ----- | ----- | ----- | ----- | ----- |
> | $R_{\mathsf{nat}}$       | 5.37  | 6.32  | 8.28  | 9.40  | 10.11 | 11.12 | 11.59 | 12.28 | 13.03 | 13.65 | 14.21 | 14.61 |
> | $\hat{R}_{\mathsf{adv}}$ | 23.37 | 13.39 | 11.87 | 11.16 | 9.65  | 8.99  | 8.82  | 8.76  | 8.01  | 8.04  | 7.86  | 7.74  |
>
> For THRM:
>
> | $\lambda$                | 1     | 10    | 50    | 100   | 150   | 200   | 250   | 300   | 400   | 600   | 800   | 1000  |
> | ------------------------ | ----- | ----- | ----- | ----- | ----- | ----- | ----- | ----- | ----- | ----- | ----- | ----- |
> | $R_{\mathsf{nat}}$       | 4.68  | 5.35  | 5.76  | 6.56  | 7.19  | 8.27  | 8.80  | 9.42  | 10.24 | 11.67 | 12.90 | 14.00 |
> | $\hat{R}_{\mathsf{adv}}$ | 55.46 | 27.53 | 18.86 | 15.10 | 13.98 | 13.05 | 12.37 | 11.87 | 11.27 | 10.94 | 10.50 | 10.42 |
>
> One interesting finding is that models trained with THRM manifest noteworthy adversarial robustness, especially on CIFAR-10, although there is no adversarial risk in the objective function of THRM. These facts also support the hypothesis (i.e., the potential mutual benefits between adversarial robustness and hypocritical robustness).
>
> **Question 7:** Shouldn't the experiment be done with larger and more variety of datasets?
>
> It is worth noting that we did exhaustive experiments to uncover the tradeoffs and relative merits of THRM and TRADES. The experiments in this paper are extraordinarily expensive. It takes over 230 GPU days on NVIDIA Tesla V100 GPU to train the models considered in this paper on CIFAR-10. We believe that the phenomena (e.g., there may be mutual benefits between adversarial robustness and hypocritical robustness) showed by our experiments are beneficial to the adversarial ML community and deserve further rigorous analysis in future work. We have clarified this in the revision. We expect that the phenomena on CIFAR-10 would generalize to larger datasets such as ImageNet, but executing the same scale experiments on ImageNet is beyond the computation cost we can afford in terms of time and money.
>
> **Question 8:** A lot of the technically machinery is not very novel with same attack as before.
>
> As discussed in Question 1, maybe the technical machinery of the hypocritical attack is not very novel to you, but we believe that the attack optimization algorithm is technically sound, and importantly, the attack objective is far overlooked by the community.
>
> **Question 9:** As a writing style, false friends is a good analogy but please do not use it throughout the paper in place of technical terms such as this line "Better transferability is beneficial for black-box attacks but is not always desired by false friends".
>
> Thanks for the correction. We have fixed them in the revision.

---

> ### Author Response · Authors · 2020-11-18
> **Response to AnonReviewer3 (1/2)**
>
> Thank you for your time and valuable comments. Below we address specific questions that you asked. We are happy to discuss and clarify further if you would find it helpful.
>
> **Question 1:** It is not very surprising that mis-classified data points can be perturbed to be properly classified using exactly the same kind of attack as prior work (not sure why authors call the attack conceptually different).
>
> Yes, it is not very surprising considering the excessive sensitivity of a model, but the phenomenon is far overlooked by the community. As far as we know, the objective of targeting a misclassified example to its correct class has never been considered in the literature. It is a reverse version of the original adversarial attack, and thus deserves a new name. This can be a problem since it can make people satisfy the model performance. We further exemplify the attack scenario by adding a concrete example in the revision. Demonstrating this potential security risk is one of the main contributions in this paper.
>
> **Question 2:** I think the work is missing an important comparison to R_bdy (and R_rob) in Zhang et al. Is that in Proposition 2? Why call it R_rev?
>
> No, the $R_{\mathsf{rev}}$ in Proposition 2 is not the $R_{\mathsf{bdy}}$ or $R_{\mathsf{rob}}$ in [1]. Their $R_{\mathsf{bdy}}$ looks similar to our two sided errors, but is actually one side error stemmed from the $R_{\mathsf{rob}}$ ($R_{\mathsf{rob}}$  is equivalent to $R_{\mathsf{adv}}$ in our paper). Specifically, $R_{\mathsf{rev}} = R_{\mathsf{bdy}}+ R_{\mathsf{nat}}$ $\overline{\overline{R}}_{\mathsf{hyp}}$.
>
> We call it $R_{\mathsf{rev}}$ because minimizing it pursues the model whose predictions can't be reversed by small perturbations.
>
> [1] Zhang, Jingfeng, et al. "Attacks Which Do Not Kill Training Make Adversarial Learning Stronger." ICML 2020.
>
> **Question 3:** Why is equation (4) a natural risk to minimize? I do not see how the tradeoff arises here.
>
> Because, at the end of Section 2.1, we noticed that it's not enough to blindly pursue robustness against hypocritical perturbations but ignore the performance on clean examples. There exists an inherent tension between the goal of natural risk minimization and hypocritical risk minimization. Thus the objective of model selection becomes to find the model which has better tradeoff between natural and hypocritical risks.
>
> To make this more clear, we move the toy example from Section 4.1 to Section 3, as another motivation that illustrates the fact that there may exist a trade-off between natural risk and hypocritical risk. Thanks for your feedback!
>
> **Question 4:** The fact that TRADES would help here is obvious, isn't it? As TRADES handles two sided errors, it naturally also takes care of the one side of the error the authors care about.
>
> Maybe it's obvious to you, but not to everyone. It's our rigorous analysis that proves, in multi-class classification settings, the relation between hypocritical risk and the objective behind TRADES. This is actually one of the contributions in this paper.
>
> **Question 5:** And that naturally raises the question that will only caring about one side of the error (in Eq 4) increase the other side (R_adv)?
>
> Thanks for your suggestion. Despite our experiments showed that there are mutual benefits between adversarial robustness and hypocritical robustness in real-world datasets, we note that in general, this synergism does not necessarily exist. In our revision of the paper, we have added Appendix D to quantitatively illustrate the trade-off between adversarial risk and hypocritical risk on another toy example.

---

### Official Review · AnonReviewer1 · 2020-10-28
**A technique to alter test images so as to mislead classifiers into self-complacency**

**Rating:** 3
**Confidence:** 3

**Review:**

1. The premise of the paper is that the adversary can perturb the *test* set so that the model is shown to perform better that it really is capable of. And in Section 7 (Conclusion) the paper claims that it exposes this new risk. However, remember that this risk is already mitigated in practice by keeping the test data *independent* of the model/classifier (e.g., see Kaggle competitions where the test set is hidden). Therefore, the perceived risk is not even present. In the context that the technique has been introduced, it seems like the [malicious] actor would only be fooling him/her self rather than fooling the model/classifier.

It is hard to come up with an application for manipulating the *test* data in the manner proposed and I am curious to hear what the authors feel.

2. What if instead of manipulating the images to conform to the classifier's predictions, we manipulate the labels, i.e., relabel the test data as per the classifier's predictions? Would that result in the same final objective being achieved?

3. Equations 1 and 2: The technique is based on generating adversarial training examples and does not make any fundamental technical contributions.

4. Abstract: "Extensive experiments verify the theoretical results and the effectiveness of our proposed methods." -- Just a couple of standard simple image datasets (MNIST, CIFAR-10) have been employed. Harder and more number of datasets should be employed in order to justify the statement in the Abstract.

5. Section 6 Discussion: "What we can do at present is using a relatively more robust model as a surrogate of the true friend to improve the robustness of a weak model." -- It is not clear how that would help. The attacker might mislead the 'relatively more robust model' as well. Moreover, why not use the 'relatively more robust model' instead of the weaker model?

One simple mitigation strategy could be to digitally sign the test set to check for tampering. Why has that not been suggested?

---

> ### Author Response · Authors · 2020-11-18
> **Response to AnonReviewer1 (2/2)**
>
> **Question 6:** Why not use the 'relatively more robust model' instead of the weaker model?
>
> That's the interesting part. There are many scenarios that we can't directly use the high-performance models to do predictions. For example, the model owner does not want to expose the model weights or the model's output directly, for the purpose of privacy protection. As another example, the output space of the high-performance model is not aligned with the output space of the weak model (e.g., the high-performance model is trained on CIFAR-10, and the weak model that needs help is trained on CIFAR-100).
>
> In such cases, to transfer knowledge at test time, the model owner can manipulate an input example to decrease the hardness for prediction from the example (this procedure can be viewed as data preprocessing at test time). Then expecting that the easiness of the manipulated example will transfer to the weak model, so that the weak model gets helped. We leave this to future work.
>
> **Question 7:** One simple mitigation strategy could be to digitally sign the test set to check for tampering. Why has that not been suggested?
>
> We implicitly assume that the test data can be stealthily perturbed by the attacker, as long as the perturbations do not change the oracle's output (i.e., human labeler). The same assumption has been made in generating adversarial examples.

---

> ### Author Response · Authors · 2020-11-18
> **Response to AnonReviewer1 (1/2)**
>
> Thank you for your time and valuable comments. Below we address specific questions that you asked. We are happy to discuss and clarify further if you would find it helpful.
>
> **Question 1:** It is hard to come up with an application for manipulating the *test* data in the manner proposed and I am curious to hear what the authors feel.
>
> We apologize that we didn't exemplify the attack scenario. Here we give an example. For instance, before allowing autonomous vehicles to drive on public roads, manufacturers must first pass tests in specific environments (closed or open roads) to obtain a license. An attacker may add imperceptible perturbations on the test examples (e.g., the "stop sign" on the road) stealthily without human notice, to hypocritically help an ML-based autonomous vehicle to pass the tests that might otherwise fail. However, the high performance can not be maintained on public roads without the help of the attacker. Thus, the potential risk is underestimated and traffic accidents might happen unexpectedly when the vehicle driving on public roads.
>
> **Question 2:** What if instead of manipulating the images to conform to the classifier's predictions, we manipulate the labels, i.e., relabel the test data as per the classifier's predictions? Would that result in the same final objective being achieved?
>
> Yes, manipulating the labels instead of perturbing the examples would result in perfect accuracy. However, the final objective might not be achieved. On the one side, such label manipulations could be easily noticed by human inspection. Practitioners might not trust the performance evaluated on a *mislabeled test set*. On the contrary, minor perturbations on clean examples possess a certain of concealment, i.e., the hypocritically perturbed test set is *correctly labeled* and looks no different from the clean counterpart but can stealthily upgrade model performance. On the other side, such label manipulations may be impractical in some scenarios. Taking the test environments (closed or open roads) of autonomous vehicles for example, if a "stop sign" is misclassified as "120km/hr", then the autonomous vehicle might accelerate to some accidents. Thus the autonomous vehicle fails the tests.
>
> **Question 3:** Equations 1 and 2: The technique is based on generating adversarial training examples and does not make any fundamental technical contributions.
>
> As far as we know, the objective of targeting a misclassified example to its correct class has never been considered in the literature. It is a reverse version of the original adversarial attack. This can be a problem since it can make people satisfy the model performance. Demonstrating this critical yet far overlooked issue is one of the main contributions in this paper. It is just a natural choice to minimize the objective using projected gradient descent.
>
> **Question 4:** Just a couple of standard simple image datasets (MNIST, CIFAR-10) have been employed. Harder and more number of datasets should be employed in order to justify the statement in the Abstract.
>
> It is worth noting that we did exhaustive experiments to uncover the tradeoffs and relative merits of THRM and TRADES. The experiments in this paper are extraordinarily expensive. It takes over 230 GPU days on NVIDIA Tesla V100 GPU to train the models considered in this paper on CIFAR-10. We believe that the phenomena (e.g., there may be mutual benefits between adversarial robustness and hypocritical robustness) showed by our experiments are beneficial to the adversarial ML community and deserve further rigorous analysis in future work. We have added these details in the revision. We expect that the phenomena on CIFAR-10 would generalize to larger datasets such as ImageNet, but executing the same scale experiments on ImageNet is beyond the computation cost we can afford in terms of time and money.
>
> **Question 5:** It is not clear how that would help. The attacker might mislead the 'relatively more robust model' as well.
>
> Yes, the 'relatively more robust model' might also be fooled. However, as long as the 'relatively more robust model' possesses better adversarial robustness, the performance of the 'weak model' would be improved with the help of the 'relatively more robust model'.

---

> > ### Comment · AnonReviewer1 · 2020-11-22
> > **No a convincing example of application**
> >
> > The autonomous vehicle example might be a simple *application* of perturbing data using adversarial techniques, but is not much else (in my opinion).
> >
> > It also does not address my concern that the test data is independent. Having access to it beforehand so as to be able to manipulate is a case of fraud in the context of passing certifications/regulations. I do not find the setup convincing at all.

---

> > > ### Author Response · Authors · 2020-11-23
> > > **An attacker is capable of manipulating examples at test time**
> > >
> > > Thanks for your active discussions.
> > >
> > > Considering you agreed that the autonomous vehicle example is indeed a simple *application* of perturbing data in the manner proposed in this paper, we believe that Question 1 was appropriately addressed. Please note that the "adversarial techniques" (hypocritical perturbations) considered in this paper have never been concerned in the literature. To the best of our knowledge, this is the first attempt to formalize the realistic risk raised by hypocritical examples.
> > >
> > > As we discussed in the answer to Question 7, it is our basic assumption that **the attackers (adversaries and false friends) are capable of manipulating examples at test time**.
> > > To achieve their evil goals, the attackers will do whatever they can, even crime. That's why they are dangerous.
> > > That's why we assumed that the test data can be stealthily perturbed by the attacker, as long as the perturbations do not change the oracle's output (i.e., human labeler). The same assumption has been made in generating adversarial examples.
> > >
> > > We sincerely hope that these facts relieve your concerns.

---

### Official Review · AnonReviewer2 · 2020-10-30
**Recommendation to reject**

**Rating:** 4
**Confidence:** 4

**Review:**

#### Problem Statement

The research on adversarial examples typically focuses on labels which are correct but non-robust: eg. an image of a bus that is classified correctly but with a few changes to pixels can be misclassified. This paper suggests that we should also pay attention to robustness in misclassification. If a classifier is not robust, even on images where it makes a mistake, changing a few pixels might fool the model into giving it the correct label, but presumably  for the wrong reasons.

The authors envision a scenario where a "false friend" examines the output of a network, and attacks the mislabeled examples to make it look like they are labeled correctly. In such a scenario, not much training happens even though the network is far from optimal on noiseless examples. They suggest certain mitigations against such attacks by favoring networks whose labels on mistakes are robust.

They propose a new objective function called THM and compare it to TRADES.

#### Pros

The paper makes a good point that (non)robustness for classifications sometimes goes hand in hand with (non)robustness for misclassifications.

#### Cons

* The attack scenario which the paper envisions does not strike me as very realistic.

* From what I can tell, TRADES would aim for robustness on all examples, whether rightly or wrongly misclassified, whereas THRM seems to focus only on robustness for misclassified instances.    It appears that adversarial risk on correctly classified examples is not being weighted. This might be a better choice for the task of avoiding the type of attacks proposed in this paper,  but overall I am not convinced it is a better objective than THRM, which aims for robustness everywhere.

* One could imagine that in the real world, when one is not concerned about false friend attacks, one would like robustness for the correct labels, but perhaps not so much for the wrong labels, as this could be a sign that the network is unsure of its prediction, and could change its output, given more data or training.

#### Summary of recommendation

The observation that non-robustness applies to correct and wrong labels is interesting. But I didn't find the attack scenario built on top of this observation and the proposed mitigations compelling.

---

> ### Author Response · Authors · 2020-11-18
> **Response to AnonReviewer2**
>
> Thank you for your time and valuable comments. Below we address specific questions that you asked. We are happy to discuss and clarify further if you would find it helpful.
>
> **Question 1:** The attack scenario which the paper envisions does not strike me as very realistic.
>
> We apologize that we didn't exemplify the attack scenario. Here we give an example. For instance, before allowing autonomous vehicles to drive on public roads, manufacturers must first pass tests in specific environments (closed or open roads) to obtain a license. An attacker may add imperceptible perturbations on the test examples (e.g., the "stop sign" on the road) stealthily without human notice, to hypocritically help an ML-based autonomous vehicle to pass the tests that might otherwise fail. However, the high performance can not be maintained on public roads without the help of the attacker. Thus, the potential risk is underestimated and traffic accidents might happen unexpectedly when the vehicle driving on public roads.
>
> **Question 2**: This might be a better choice for the task of avoiding the type of attacks proposed in this paper, but overall I am not convinced it is a better objective than TRADES, which aims for robustness everywhere.
>
> Our analysis just shows that THRM is optimizing a tight upper bound of hypocritical risk than TRADES. This doesn't mean that THRM will always be better. In practice, the superiority depends on their actual performance on adversarial robustness and/or hypocritical robustness. Their performance changes with specific problems and optimization algorithms.
>
> We agree that, in general, robustness on both correctly classified and misclassified examples is an appealing property. At the same time, our experiments show that there may be mutual benefits between adversarial robustness and hypocritical robustness. This is an intriguing synergism phenomenon that deserves further rigorous analysis in future work.
>
> **Question 3:** One could imagine that in the real world, when one is not concerned about false friend attacks, one would like robustness for the correct labels, but perhaps not so much for the wrong labels, as this could be a sign that the network is unsure of its prediction.
>
> Yes, unstable model prediction on misclassified examples indeed could be a sign that the network is unsure of its prediction. But also, unstable prediction on correctly classified examples is another sign. That's why (non)robustness for classifications sometimes goes hand in hand with (non)robustness for misclassifications, and we should pay attention to both of them.
>
> **Question 4:** I didn't find the attack scenario built on top of this observation and the proposed mitigations compelling.
>
> The attack scenario is explained in Question 1. The mitigations suggested in the paper (THRM and TRADES) are based on an initial analysis of hypocritical risk. It is our theoretical analysis that justifies that TRADES is optimizing an upper bound of the hypocritical risk, although it is originally designed for adversarial robustness. This is actually one of the contributions in this paper. We sincerely hope that this fact relieves your concerns.

---

### Official Review · AnonReviewer4 · 2020-11-02
**I am waiting for the author's response.**

**Rating:** 7
**Confidence:** 3

**Review:**

Summary
This paper presents a new kind of adversarial attacks, named hypocritical attack. It is a reverse version of the original adversarial attack. It tricks a model into classifying data correctly with a perturbation. This can be a problem since it can make people satisfy the model performance, but the model is not robust on the real test dataset. The authors review the adversarial attack and define the new hypocritical examples and risk. The authors also show the simple results why hypocritical attach is a critical issue by a Naive model that is initialized randomly. It shows high performance on the hypocritical examples but low on the clean test data. They also investigate the algorithms that improve model robustness, THRM, and TRADES. Experiments show a trade-off between original classification loss and hypocritical risks, and THRM is a tight upper bound against the TRADES.

The main strength of this paper is suggesting a sound and straightforward attack named hypocritical attack. I think almost machine learning (ML) researchers cheer with joy when they see the high-performance results on their model. But sometimes, we need to think about our mistakes or test data characteristics (i.e., easy test data). This paper suggests another consideration to the ML community. The authors also investigate minimization algorithms on the attack risk and show future directions such as transferability.

I have several questions about this paper that I want to listen to the author's response.


Questions and Comments
Is there any trade-off between adversarial risk and hypocritical risk like precision-recall trade-off?
What if we apply the technique to make hypocritical examples in the training data? We can measure the training accuracy then we can make the examples from the training data. Do these examples boost the classifier?
It would be better to describe the toy example since it is hard to understand the text's experiment in this paper. Even this experiment is borrowed from Zhang et al. 2019, readers want to understand the experiment with this paper only.
Typo: with large hypocritical hypocritical risk, while the optimal on page 7.

---

> ### Author Response · Authors · 2020-11-18
> **Response to AnonReviewer4**
>
> Thank you for the supportive review and kind suggestions. Below we address specific questions that you asked. We are happy to discuss and clarify further if you would find it helpful.
>
> **Question 1:** Is there any trade-off between adversarial risk and hypocritical risk like precision-recall trade-off?
>
> Thanks for pointing this out. Despite our experiments showed that there are mutual benefits between adversarial robustness and hypocritical robustness on CIFAR-10, we note that in general, this synergism does not necessarily exist. In our revision of the paper, we have added Appendix D to quantitatively illustrate the trade-off between adversarial risk and hypocritical risk on another toy example.
>
> **Question 2:** We can measure the training accuracy then we can make the examples from the training data. Do these examples boost the classifier?
>
> Yes. We could construct such a test set, which only contains correctly classified training examples. Considering that overparameterized deep networks have the capacity to memorize training data with zero training error, models will manifest like a perfect one. But this paper focuses on model robustness on natural examples: a hypocritical attacker can easily help a model to do correct classification with little perturbations.
>
> **Question 3:** It would be better to describe the toy example since it is hard to understand the text's experiment in this paper. Even this experiment is borrowed from Zhang et al. 2019, readers want to understand the experiment with this paper only.
>
> We appreciate the kind suggestion. The text's experiment is actually a motivation that illustrates the fact that there may exist a trade-off between natural risk and hypocritical risk. That's why we should optimize our models to minimize them at the same time. To make it more clear, the text's experiment has been moved to Section 3 to motivate algorithmic designs in the revision.
>
> **Question 4:** Typo: with large hypocritical hypocritical risk, while the optimal on page 7.
>
> Thanks for the correction. We have fixed it in the revision.

---

### Author Response · Authors · 2020-11-18
**Updates in Paper**

We sincerely appreciate all reviewers for their time and insightful comments.

We have uploaded the manuscript based on valuable conversations with all reviewers. The suggested modifications in the revised manuscript are highlighted in blue. Here we briefly summarize them:

1. We have added two statements in the introduction section to enhance our arguments on hypocritical examples.
   - It is of scientific interest to study hypocritical examples--the opposite of adversarial examples.
   - There are practical threats. We exemplify the attack scenario with a concrete example of autonomous vehicles.
2. We have moved the toy example from Section 4.1 to Section 3, to serve as a motivation that illustrates the fact that there may exist a trade-off between natural risk and hypocritical risk. Then we combine the remainder of Section 4 with Section 5 as a single section.
3. We have provided the details about computation cost in Section 4.
4. We have added evaluation results about $\hat{R}_{\mathsf{adv}}$ at the end of Appendix A.3.
5. We have included Appendix D to quantitatively illustrate the trade-off between adversarial risk and hypocritical risk on another toy example.

---

### Decision · Program_Chairs · 2021-01-07
**Final Decision**

**Decision:**

Reject

**Comment:**

This paper the flip-side of an adversarial "attack" in that data may be perturbed to make it look like a model was performing well rather than the standard notion of adversarial attacks. The reviewers found this notion interesting and potentially worthy of investigation. However as it stands, the proposed applications and methods do not seem developed well enough as would be expected at a conference like this.